# SegRefiner: Towards Model-Agnostic Segmentation Refinement with Discrete Diffusion Process

**Mengyu Wang**[1,3][*], **Henghui Ding**[4], **Jun Hao Liew**[5], **Jiajun Liu**[5]
**Yao Zhao**[1,2,3][†], **Yunchao Wei**[1,2,3][†]

[1] Institute of Information Science, Beijing Jiaotong University
[2] Peng Cheng Laboratory
[3] Beijing Key Laboratory of Advanced Information Science and Network
[4] Nanyang Technological University    [5] ByteDance
wmengyu0826@gmail.com

## Abstract

In this paper, we explore a principal way to enhance the quality of object masks produced by different segmentation models. We propose a model-agnostic solution called SegRefiner, which offers a novel perspective on this problem by interpreting segmentation refinement as a data generation process. As a result, the refinement process can be smoothly implemented through a series of denoising diffusion steps. Specifically, SegRefiner takes coarse masks as inputs and refines them using a discrete diffusion process. By predicting the label and corresponding states-transition probabilities for each pixel, SegRefiner progressively refines the noisy masks in a conditional denoising manner. To assess the effectiveness of SegRefiner, we conduct comprehensive experiments on various segmentation tasks, including semantic segmentation, instance segmentation, and dichotomous image segmentation. The results demonstrate the superiority of our SegRefiner from multiple aspects. Firstly, it consistently improves both the segmentation metrics and boundary metrics across different types of coarse masks. Secondly, it outperforms previous model-agnostic refinement methods by a significant margin. Lastly, it exhibits a strong capability to capture extremely fine details when refining high-resolution images. The source code and trained models are available at github.com/MengyuWang826/SegRefiner.

## 1 Introduction

Although segmentation in image [35, 23, 24, 14, 11] and video [54, 37, 38, 16, 15] has been extensively studied in the past decades, obtaining accurate and detailed segmentation masks is always challenging since high-quality segmentation requires the model to capture both high-level semantic information and fine-grained texture information to make accurate predictions. This challenge is particularly pronounced for images with resolutions of *2K* or higher, which requires considerable computational complexity and memory usage. As a result, existing segmentation algorithms often predict masks at a smaller size, inevitably leading to lower accuracy due to the loss of fine-grained information during downsampling.

Since directly predicting high-quality masks is challenging, some previous works have shifted their attention to the refinement of coarse masks obtained from a preceding segmentation model. A popular line of direction [29, 57, 27, 28] is to augment the segmentation models (and features) with a new

---

[*]Work done during an internship at ByteDance.
[†]Co-corresponding author.

37th Conference on Neural Information Processing Systems (NeurIPS 2023).

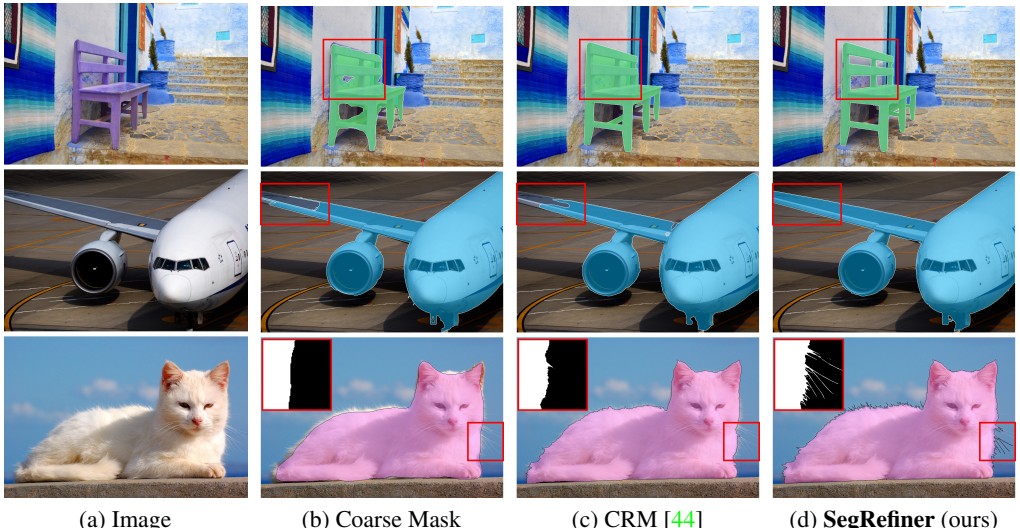

| (a) Image | (b) Coarse Mask | (c) CRM [44] | (d) **SegRefiner** (ours) |

Figure 1: Diverse errors presented in the previous segmentation results. The top row and middle row show the significant false positives and false negatives caused by incorrect semantics, the bottom row shows the inaccurate in capturing fine-grained details. Please zoom in for a better view.

module for masks correction. However, such approaches are usually model-specific, and hence cannot be generalized to refine coarse masks produced by other segmentation models. Another part segmentation refinement works [47, 56, 12, 44], on the other hand, resort to model-agnostic approach by taking only an image and the coarse segmentation as input for refinement. These methods have greater practical utility, as they are applicable to refine different segmentation models. However, the diverse types of errors (*e.g.*, errors along object boundaries, failure in capturing fine-grained details in high-resolution images and errors due to incorrect semantics) presented in the coarse masks pose a great challenge to the refinement model thus causing underperformance (refer to Fig. 1).

In response to this challenge, we draw inspiration from the working principle of denoising diffusion models [45, 22, 2]. Diffusion models perform denoising at each timestep and gradually approach the image distribution through multiple iterations. This iterative strategy significantly reduces the difficulty of fitting the target distribution all at once, empowering diffusion models to generate high-quality images. Intuitively, by applying this strategy to the segmentation refinement task, refinement model can focus on correcting only some "most obvious errors" at each step and iteratively converge to an accurate result, thus reducing the difficulty of correcting all errors in a single pass and enabling refinement model to handle more challenging instances.

Under this perspective, we present an innovative interpretation of the task by representing segmentation refinement as a data generation process. As a result, refinement can be implemented through a sequence of denoising diffusion steps with coarse segmentation masks being the noisy version of ground truth. To work with binary masks, we further devise a novel discrete diffusion process, where every pixel performs *unidirectional randomly states-transition*. The proposed process can gradually convert ground truth into a coarse mask during training and employ the coarse mask as sampling origination during inference. In other words, we formulate the mask refinement task as a conditional generation problem, where the input image serves as the condition for iteratively updating/refining the erroneous predictions in the coarse mask.

To our best knowledge, we are the first to introduce diffusion-based refinement for segmentation masks. Our method, called SegRefiner, is model-agnostic, and thus applicable across different segmentation models and tasks. We extensively analyze the performance of SegRefiner across various segmentation tasks, demonstrating that our SegRefiner not only outperforms all previous model-agnostic refinement methods (with +3.42 IoU, +2.21 mBA in semantic segmentation and +0.9 Mask AP, +2.2 Boundary AP in instance segmentation), but can also be effortlessly transferred to other segmentation tasks (*e.g.*, the recently proposed dichotomous image segmentation task [40]) without any modification. Additionally, SegRefiner demonstrates a strong capability for capturing extremely fine details when applied to high-resolution images (see Fig. 1 and Fig. 4).

## 2 Related Work

**Segmentation Refinemement**   The aim of segmentation refinement is to improve the quality of masks in pre-existing segmentation models. Some works focus on enhancing specific segmentation models. PointRend [29] employs an MLP to predict the labels of pixels with low-confidence scores output from Mask R-CNN [21]. RefineMask [57] incorporates a semantic head to Mask R-CNN as additional guidance. MaskTransfiner [27] employs an independent FCN [35] to detect incoherent regions and refines their labels with a Transformer [49]. These works have demonstrated notable performance enhancements over their preceding models. However, their scope of improvement is model-specific and they lack the capacity to directly refine the coarse mask derived from other models. There are also some refinement methods that adopt model-agnostic approaches, such as [47, 56, 61, 12, 44, 31]. These strategies emphasize utilizing diverse forms of input, including whole images, boundary patches, edge strips, *etc.* Even though these techniques can refine coarse masks derived from different models, their applicability remains confined to specific segmentation tasks. As a special case, SegFix [56], which learns a mapping function between edge pixels and inner pixels and subsequently replaces inaccurate edge predictions with corresponding inner pixel predictions, is employed in both semantic segmentation and instance segmentation within the Cityscapes dataset [13]. While the performance of SegFix is significantly constrained by its ability to accurately identify objects within an image, which consequently leads to a decline in performance on datasets with a more extensive range of categories (*e.g.*, COCO [34]).

**Diffusion Models for Detection and Segmentation**   Recently, diffusion models have received a lot of attention in research. Initial studies [45, 22, 2, 46, 8, 3] primarily sought to enhance and expand the diffusion framework. Following these, subsequent works ventured to incorporate diffusion models across a broader array of tasks [43, 9, 30] and to formulate comprehensive conditional generation frameworks [41, 58]. The application of diffusion models to detection and segmentation tasks has also been the focus of an escalating number of studies. Baranchuk *et al.* [4] capture the intermediate activations from diffusion models and employ an MLP to execute per-pixel classification. DiffusionDet [6] and DiffusionInst [19] adapt the diffusion process to perform denoising in object boxes and mask filters. Some other works [1, 52, 53, 7, 26] consider the image segmentation task as mask generation. These studies predominantly follow the Gaussian diffusion process of DDPM [22] and leverage an additional image encoder to extract image features as condition to generate masks. To the best of our knowledge, our SegRefiner is the first work that applies a diffusion model to the image segmentation refinement task, and it also pioneers the abandonment of the Gaussian assumption in favor of a newly designed discrete diffusion process in diffusion-based segmentation tasks.

## 3 Methodology

### 3.1 Preliminaries: Diffusion Models

Diffusion models consist of a forward and a reverse process. The forward process $q(x_{1:T}|x_0)$ uses a Markov or None-Markov chain to gradually convert the data distribution $x_0 \sim q(x_0)$ into complete noise $x_T$ whereas the reverse process deploys a gradual denoising procedure $p_\theta(x_{0:T})$ that transforms a random noise back into the original data distribution.

**Continuous Diffusion Models**   The majority of existing continuous diffusion models [22, 46, 41] adheres to the Gaussian assumption and defines $p(x_T) = \mathcal{N}(x_T|0, 1)$. The mean and variance of forward process are defined by a hyperparameter $\beta_t$ and the reverse process utilizes mean and variance from model predictions, thus formulating as:

$$q(x_t|x_{t-1}) = \mathcal{N}(x_t|\sqrt{1-\beta_t}x_{t-1}, \beta_t I), \tag{1}$$

$$p_\theta(x_{t-1}|x_t) = \mathcal{N}(x_{t-1}|\mu_\theta(x_t, t), \Sigma_\theta(x_t, t)). \tag{2}$$

**Discrete Diffusion Models**   Compared to continuous diffusion models, there is less research on discrete diffusion models. Sohl-Dickstein *et al.* [45] first introduce binary diffusion to reconstruct one-dimensional noisy binary sequences. $x_T$ is defined to adhere to the Bernoulli distribution $\mathcal{B}(x_T|0.5)$. The forward process and reverse process are represented as:

$$q(x_t|x_{t-1}) = \mathcal{B}(x_t|x_{t-1}(1-\beta_t) + 0.5\beta_t), \tag{3}$$

$$p_\theta(x_{t-1}|x_t) = \mathcal{B}(x_{t-1}|f_b(x_t, t)). \tag{4}$$

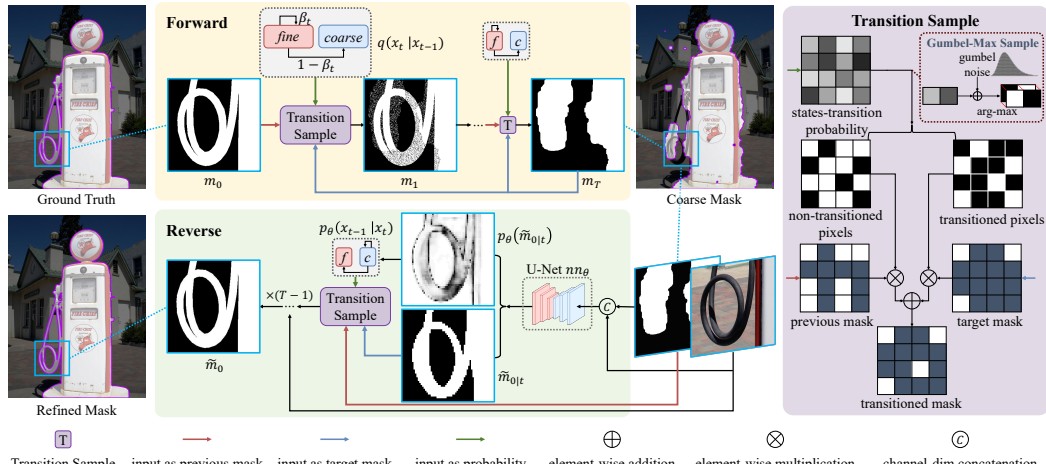

Figure 2: **An overview of the proposed SegRefiner** (best viewed in color). On the right is the transition sample module we proposed, which randomly samples pixels from the current mask based on the input states-transition probabilities and change their values to match those in the target mask. During training, the transition sample module transforms the ground truth into a coarse mask, thus coarse mask is the target mask. During inference, target mask refers to the predicted fine mask and this module updates the values in the coarse mask in each timestep based on the predicted fine mask and the transition probabilities.

where $\beta_t \in (0, 1)$ is a hyperparameter and $f_b(x_t, t)$ is a model predicting Bernoulli probability. After the great success of DDPM [22], Austin *et al.* [2] extended the architecture of Discrete Diffusion Model to a more general form. They define the forward process as a discrete random variable transitioning among multiple states and use states-transition matrix $Q_t$ to characterize this process:

$$[Q_t]_{m,n} = q(x_t = n | x_{t-1} = m). \tag{5}$$

## 3.2  SegRefiner

In this work, we propose SegRefiner, with a unique discrete diffusion process, which can be applied to refine coarse masks from various segmentation models and tasks. SegRefiner performs refinement with a *coarse-to-fine* diffusion process. In the forward process, SegRefiner employs a discrete diffusion process which is formulated as unidirectional random states-transition, gradually degrading the ground truth mask into a coarse mask. In the reverse process, SegRefiner begins with a provided coarse mask and gradually transforms the pixels in the coarse mask to the refined state, correcting the wrongly predicted area in the coarse mask. In the following paragraphs, we will provide a detailed description of the forward and reverse process.

**Forward diffusion process**   In the forward process, we gradually degrade the ground truth mask/ fine mask $M_{fine}$, transiting it into a coarse mask $M_{coarse}$. In other words, we have $m_0 = M_{fine}$ and $m_T = M_{coarse}$. At any intermediate timestep $t \in \{1, 2, ..., T-1\}$, the intermediate mask $m_t$ is therefore in a transitional phase between $M_{fine}$ and $M_{coarse}$.

We define that every pixel in $m_t$ has two states: fine and coarse, and the forward process is thus formulated as states-transition between these two states. Pixels in the fine state will retain their values from $M_{fine}$, and *vice versa*. We propose a new transition sample module to formulate this process. As shown in Fig. 2 right, during forward process, the transition sample module takes the previous mask $m_{t-1}$, coarse mask $m_T$ and a states-transition probability as input and outputs a transitioned mask $m_t$. The states-transition probability describes the probability of every pixel in $m_{t-1}$ transitioning to the coarse state. This module first performs Gumbel-max sampling [25] according to the states-transition probability and obtains the transitioned pixels. Then, the transitioned pixels will take values from $m_T$ while the non-transitioned pixels will keep unchanged.

Note that the transition sample module represents a **unidirectional** process in which only "transition to coarse state" happens. The unidirectional property ensures that the forward process will converges

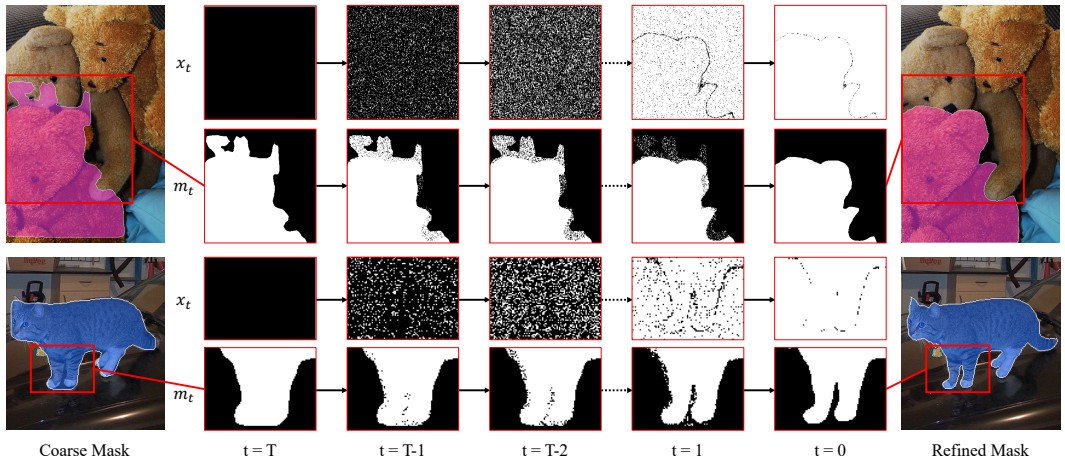

| Coarse Mask | t = T | t = T-1 | t = T-2 | t = 1 | t = 0 | Refined Mask |

Figure 3: **Examples of SegRefiner's inference process.** $x_t$ and $m_t$ denote the state (we represent state $[0, 1]$ and $[1, 0]$ as 0 and 1 in this figure) and the corresponding mask, respectively. SegRefiner begins with $x_T = [0, 1]$ and $m_T = M_{coarse}$, and gradually refines until we obtain fine mask $m_0$.

to $M_{coarse}$, despite each step is completely random. This is a significant difference between our SegRefiner and previous diffusion models in which forward process converges to a random noise.

With the reparameterization trick, we introduce a binary random variable $x$ to formulate the above process. We represent $x_t^{i,j}$ as a one-hot vector to represent the state of pixel $(i, j)$ in $m_t$ and set $x_0^{i,j} = [1, 0]$ and $x_T^{i,j} = [0, 1]$ to represent the fine state and coarse state, respectively. The forward process can thus be formulated as:

$$q(x_t^{i,j}|x_{t-1}^{i,j}) = x_{t-1}^{i,j}Q_t, \quad \text{where } Q_t = \begin{bmatrix} \beta_t & 1 - \beta_t \\ 0 & 1 \end{bmatrix}, \tag{6}$$

where $\beta_t \in [0, 1]$, and $1 - \beta_t$ corresponds to the states-transition probability used in our transition sample module. $Q_t$ is a states-transition matrix. The form of $Q_t$ explicitly manifests the unidirectional property, *i.e.*, all pixels in the coarse state will never transition back to the fine state since $q(x_t|[0, 1]) = [0, 1]$. According to Eq. (6), the marginal distribution can be formulated as

$$q(x_t^{i,j}|x_0^{i,j}) = x_0^{i,j}Q_1Q_2 \dots Q_t = x_0\bar{Q}_t = x_0 \begin{bmatrix} \bar{\beta}_t & 1 - \bar{\beta}_t \\ 0 & 1 \end{bmatrix}, \tag{7}$$

where $\bar{\beta}_t = \beta_1\beta_2 \dots \beta_t$. Given this, we can obtain the intermediate mask $m_t$ at any intermediate timestep $t$ without the need of step-by-step sampling $q(x_t|x_{t-1})$, allowing faster training.

**Reverse diffusion process** The reverse diffusion process takes a coarse mask $m_T$ and gradually transforms it into a fine mask $m_0$. However, since the fine mask $m_0$ and the reversed states-transition probability is unknown, following DDPM [22], we train a neural network $f_\theta$ parameterized by $\theta$ to predict the fine mask $\tilde{m}_{0|t}$ at each timestep, represented as

$$\tilde{m}_{0|t}, \ p_\theta(\tilde{m}_{0|t}) = f_\theta(I, m_t, t), \tag{8}$$

where $I$ is the corresponding image. $\tilde{m}_{0|t}$ and $p_\theta(\tilde{m}_{0|t})$ denote the predicted binary fine mask and its confidence score, respectively. To obtain the reversed states-transition probability, according to Eq. (6), Eq. (7), and Bayes' theorem, we first formulate the posterior at timestep $t - 1$ as:

$$q(x_{t-1}|x_t, x_0) = \frac{q(x_t|x_{t-1}, x_0)q(x_{t-1}|x_0)}{q(x_t|x_0)} = \frac{x_tQ_t^\top \odot x_0\bar{Q}_{t-1}}{x_0\bar{Q}_t x_t^\top}, \tag{9}$$

where the fine state $x_0$ is set to $[1, 0]$ during training, indicating ground truth. While during inference, $x_0$ is unknown, as the predicted $\tilde{m}_{0|t}$ may not be entirely accurate. Since the confidence score $p_\theta(\tilde{m}_{0|t})$ represents the model's level of certainty for each pixel prediction being correct, $p_\theta(\tilde{m}_{0|t})$ can also be interpreted as the probability being in the fine state. Therefore, intuitively, one could

obtain the state of every pixel in $\tilde{m}_{0|t}$ by simply thresholding as done in [2]:

$$x_{0|t}^{i,j} = \begin{cases} [1,0] & \text{if } p_\theta(\tilde{m}_{0|t})^{i,j} \geq 0.5 \\ [0,1] & \text{otherwise,} \end{cases} \tag{10}$$

where pixels with higher confidence scores will have $x_{0|t}^{i,j} = [1,0]$, indicating they are in the fine state, and *vice versa*. However, in such one-hot form, the values of states-transition probabilities are determined solely by the pre-defined hyperparameters, leading to significant information loss. Instead, we retain the soft transition and formulate $x_{0|t}^{i,j} = [p_\theta(\tilde{m}_{0|t})^{i,j}, 1 - p_\theta(\tilde{m}_{0|t})^{i,j}]$. With this setting, the reverse diffusion process can be reformulated as

$$p_\theta(x_{t-1}^{i,j}|x_t^{i,j}) = x_t^{i,j} P_{\theta,t}^{i,j}, \quad \text{where } P_{\theta,t}^{i,j} = \begin{bmatrix} 1 & 0 \\ \frac{p_\theta(\tilde{m}_{0,t})^{i,j}(\bar{\beta}_{t-1}-\bar{\beta}_t)}{1-p_\theta(\tilde{m}_{0,t})^{i,j}\bar{\beta}_t} & \frac{1-p_\theta(\tilde{m}_{0,t})^{i,j}\bar{\beta}_{t-1}}{1-p_\theta(\tilde{m}_{0,t})^{i,j}\bar{\beta}_t} \end{bmatrix}, \tag{11}$$

where $P_{\theta,t}^{i,j}$ is the reversed states-transition matrix. With the above reversed states-transition probability, $m_t$ and $\tilde{m}_{0|t}$ as input, the transition sample module can transit a portion of pixels to the fine state at each timestep, thereby correcting erroneous predictions.

**Inference**  Given a coarse mask $m_T$ and its corresponding image $I$, we first initialize that all pixels are in the coarse state thus $x_T^{i,j} = [0,1]$. We iterate between: (1) forward pass to obtain $\tilde{m}_{0|t}$ and $p_\theta(\tilde{m}_0|t)$ (Eq. (8)); (2) compute the reversed states-transition matrix $P_{\theta,t}^{i,j}$ and obtain $x_{t-1}$ (Eq. (11)); (3) compute the refined mask $m_{t-1}$ based on $x_{t-1}$, $m_t$ and $\tilde{m}_{0|t}$. The process (1)-(3) is iterated until we obtain fine mask $m_0$. Visualization examples of inference are shown in Fig. 3.

## 4  Experiments

### 4.1  Implementation Details

**Model Architecture**  Following [39], we employ U-Net for our denoising network. We modify the U-Net to take in 4-channel input (concatenation of image and the corresponding mask $m_t$) and output a 1-channel refined mask. Both input and output resolution is set to 256×256. All others remain unchanged other than the aforementioned modifications.

**Objective Function**  Following [12], we employ a combination of binary cross-entropy loss and texture loss for training our model, *i.e.*, $\mathcal{L} = \mathcal{L}_{bce} + \alpha\mathcal{L}_{texture}$, where texture loss is characterized as an L1 loss between the segmentation gradient magnitudes of the predicted mask and the ground truth mask. $\alpha$ is set to 5 to balance the magnitude of both losses.

**Noise Schedule**  Theoretically, the unidirectional property of SegRefiner ensures that any noise schedule can make the forward process converge to the coarse mask given an infinite number of timesteps. However, in practice, we use much fewer timesteps ($T = 6$ this work) to ensure efficient inference. We designate $\bar{\beta}_T = 0$ such that $x_T = [0,1]$ for all pixels and $m_T = M_{coarse}$ (Eq. (7)). Following DDIM [46], we directly set a linear noise schedule from 0.8 to 0 on $\bar{\beta}_t$.

**Training Strategy**  Our SegRefiner model has been developed into two versions for refinement of different resolution images: a low-resolution variant (hereafter referred to as *LR-SegRefiner*) and a high-resolution variant (hereafter referred to as *HR-SegRefiner*). While these two versions employ different training datasets, all other settings remain consistent. *LR-SegRefiner* is trained on the LVIS dataset [20], whereas *HR-SegRefiner* is trained on a composite dataset merged from two high-resolution datasets, DIS5K [40] and ThinObject-5K [32]. These datasets were chosen due to their highly accurate pixel-level annotations, thereby facilitating the training of our model to capture fine details more effectively. Following [12], the coarse masks used for training are obtained through various morphological operations, such as randomly perturbing some edge points of the ground truth and performing dilation, erosion, *etc*. During training, we first train *LR-SegRefiner* on the low-resolution dataset until convergence. Subsequently, it is fine-tuned on the high-resolution dataset to yield *HR-SegRefiner*.

We employed *double random crop* as the primary data augmentation technique. It entails the random cropping of each image to generate two distinct model inputs: one encapsulating the entire foreground

target, and the other containing only a partial foreground. Both crops are subsequently resized to match the model's input size. This operation ensures that the model is proficient in refining both entire objects and incomplete local patches, a capability that proved instrumental in subsequent experiments. All the following experiments were conducted on 8 NVIDIA RTX 3090. For more details on the training process, please refer to the supplemental materials.

## 4.2 Semantic Segmentation

**Dataset and Metrics**    As the refinement task emphasizes the enhancement of mask quality, a dataset with high annotation quality and sufficient detailed information is required to evaluate the model's performance. Hence, we report the results on BIG dataset [12], a semantic segmentation dataset specifically designed for high-resolution images. With resolutions ranging from $2048 \times 1600$ to $5000 \times 3600$, this dataset provides a challenging testbed for evaluating refinement methods. The metrics we used are the standard segmentation metric IoU and the boundary metric mBA (mean Boundary Accuracy [12]), which is commonly used in previous refinement works.

**Settings**    Because of the high resolution of images in BIG dataset, we employ the *HR-SegRefiner* in this experiment. While the model's output size is only $256 \times 256$, which is insufficient for such a high-resolution dataset and results in the loss of many edge details. Consequently, during inference, we deploy the first $T - 1$ timesteps to perform global refinement, which takes the resized entire image as input, and the final timestep as a local step that takes original-size local patches as input. In order to identify the local patches that require refinement, we filter out pixels with low state-transition probabilities from the globally refined mask and use them as the center points for the local patches. We allocate more timesteps for global refinement since global steps need to handle more severe errors, thus presenting a higher difficulty level. The training strategy we have employed ensures that the model can adapt to both global and local input without modification.

**Results**    As shown in Tab. 1, we compare the proposed SegRefiner with three model-agnostic semantic segmentation refinement methods, SegFix [56], CascadePSP [12], and CRM [44]. Additionally, we include the fine-grained matting method, MGMatting [55], which employs an image and mask for matting and can also be utilized for refinement purposes. The proposed SegRefiner demonstrates superior performance over previous methods when using coarse masks from four different semantic segmentation models, as evident in both IoU and mBA metrics. Notably, our SegRefiner outperforms CRM, which is specifically designed for ultra-high-resolution images, showcasing significant advancements. We report the error bar ($\pm$ in gray) in this experiment due to the relatively small size of the BIG dataset (consisting of 100 testing images). The error bar represents the maximum fluctuation value among the results of five experiments, and the results in Tab. 1 are the average of these five trials. In subsequent experiments, the stability of the results is ensured by the availability of an ample number of testing images.

## 4.3 Instance Segmentation

**Dataset and Metrics**    To evaluate the effectiveness of our SegRefiner in refining instance segmentation, we select the widely-used COCO dataset [34] with LVIS [20] annotations. LVIS annotations offer superior quality and more detailed structures compared to the original COCO annotations and other commonly used instance segmentation datasets such as Cityscapes [13] and Pascal VOC [17]. This makes LVIS annotations more suitable for assessing the performance of refinement models. The evaluation metrics are the Mask AP and Boundary AP. It worth noting that these metrics are computed using the LVIS [20] annotations on the COCO validation set. The Boundary AP metric, introduced by Cheng *et al.* [10], is a valuable evaluation metric that measures the boundary quality of the predicted masks and is highly sensitive to the accuracy of edge prediction. It provides a detailed assessment of how well the refined masks capture the boundaries of the objects.

Table 1: IoU/mBA results on the BIG dataset comparing with other mask refinement methods.

| IoU/mBA | Coarse Mask | SegFix [56] | MGMatting [55] | CascadePSP [12] | CRM [44] | SegRefiner (ours) |
|---|---|---|---|---|---|---|
| FCN-8s [35] | 72.39 / 53.63 | 72.69 / 55.21 | 72.31 / 57.32 | 77.87 / 67.04 | 79.62 / 69.47 | **86.95**$_{\pm 0.06}$ / **72.81**$_{\pm 0.05}$ |
| DeepLab V3+ [5] | 89.42 / 60.25 | 89.95 / 64.34 | 90.49 / 67.48 | 92.23 / 74.59 | 91.84 / 74.96 | **94.86**$_{\pm 0.04}$ / **77.64**$_{\pm 0.05}$ |
| RefineNet [33] | 90.20 / 62.03 | 90.73 / 65.95 | 90.98 / 68.40 | 92.79 / 74.77 | 92.89 / 75.50 | **95.12**$_{\pm 0.05}$ / **76.93**$_{\pm 0.05}$ |
| PSPNet [60] | 90.49 / 59.63 | 91.01 / 63.25 | 91.62 / 66.73 | 93.93 / 75.32 | 94.18 / 76.09 | **95.30**$_{\pm 0.03}$ / **77.46**$_{\pm 0.04}$ |
| Avg Improve | 0.00 / 0.00 | 0.47 / 3.30 | 0.73 / 6.10 | 3.58 / 14.05 | 4.01 / 15.12 | **7.43 / 17.33** |

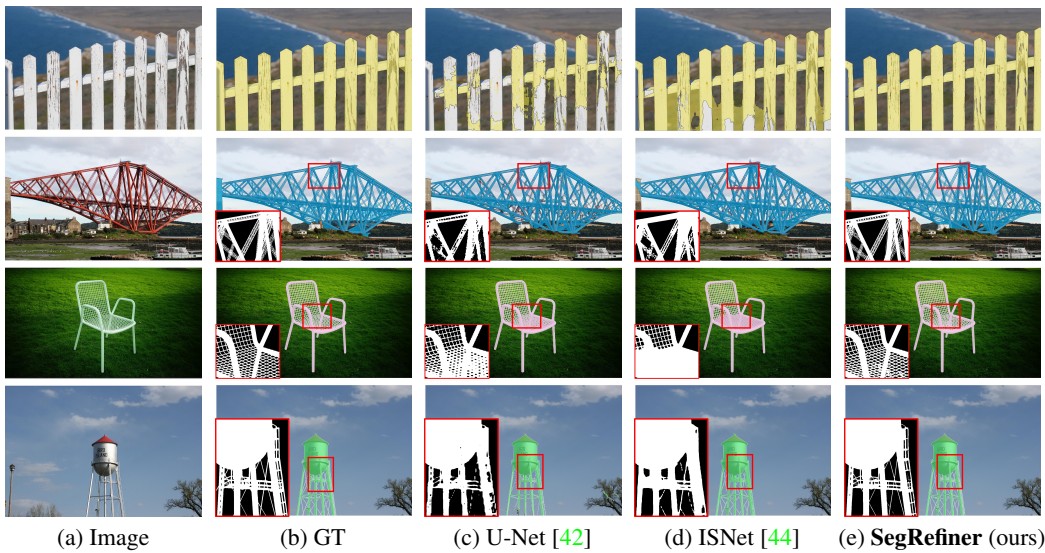

| (a) Image | (b) GT | (c) U-Net [42] | (d) ISNet [44] | (e) **SegRefiner** (ours) |

Figure 4: Qualitative comparisons with other methods on DIS5K dataset [40]. Our SegRefiner has the capability to capture finer details, allowing it to discern and incorporate more subtle nuances. Please kindly zoom in for a better view. More visual results are provided in the supplemental file.

**Settings** In this experiment, we utilize the *LR-SegRefiner* model. To refine each instance, we extract the bounding box region based on the coarse mask and expand it by 20 pixels on each side. The extracted region is then resized to match the input size of the model. The output size is suitable for instances in the COCO dataset, allowing us to perform instance-level refinement for all timesteps without requiring any local patch refinement.

**Results** First, in Tab. 2, we compare the proposed SegRefiner with two model-agnostic instance segmentation refinement methods, BPR [47] and SegFix [56]. As demonstrated in Tab. 2, our SegRefiner achieves significantly better performance compared to these two methods. The coarse masks utilized in Tab. 2 are obtained from Mask R-CNN to ensure consistency with the original experiments conducted in [47, 56]. This choice allows for a fair and direct comparison with the previous works. Then in Tab. 3, we apply our SegRefiner to other 7 instance segmentation models. Our method yields significant enhancements across models of varying performance levels. Furthermore, when compared to three model-specific instance segmentation refinement models, including PointRend [29], RefineMask [57], and Mask TransFiner [27], our SegRefiner consistently enhances their performance. These results establish SegRefiner as the leading model-agnostic refinement method for instance segmentation.

### 4.4 Dichotomous Image Segmentation

Dichotomous Image Segmentation (DIS) is a recently introduced task by Qin *et al.* [40], which specifically targets the segmentation of objects with complex texture structures, such as the steel framework of bridge illustrated in Fig. 4. To facilitate research in this area, Qin *et al.* also built the DIS5K dataset, a meticulously annotated collection of 5,470 images with resolutions of 2K and above. The DIS5K dataset, characterized by its abundance of fine-grained structural details, poses rigorous

Table 2: Comparision with other model-agnostic instance segmentation refinement methods on the COCO val set using LVIS annotations.

| Method | Mask AP | | | | | | Boundary AP | | | | | |
| | AP | AP$_{50}$ | AP$_{75}$ | AP$_S$ | AP$_M$ | AP$_L$ | AP | AP$_{50}$ | AP$_{75}$ | AP$_S$ | AP$_M$ | AP$_L$ |
|---|---|---|---|---|---|---|---|---|---|---|---|---|
| MaskRCNN (Res50) | 39.8 | 61.4 | 42.3 | 24.9 | 47.0 | 55.1 | 27.3 | 53.3 | 25.2 | 24.9 | 41.8 | 27.4 |
| + SegFix [56] | 40.6 | 61.4 | 42.8 | 25.0 | 48.4 | 56.6 | 29.1 | 53.7 | 28.0 | 24.9 | 43.6 | 30.8 |
| + BPR [47] | 41.0 | 61.4 | 43.1 | 24.8 | 48.5 | 57.8 | 30.4 | 55.2 | 29.5 | 24.7 | 43.8 | 33.7 |
| + **SegRefiner** (ours) | **41.9** | **61.6** | **43.2** | **25.7** | **49.4** | **58.8** | **32.6** | **55.7** | **32.5** | **25.6** | **45.0** | **37.3** |
| MaskRCNN (Res101) | 41.6 | 63.3 | 44.4 | 26.5 | 49.5 | 57.8 | 29.0 | 55.2 | 26.7 | 26.3 | 44.4 | 29.8 |
| + SegFix [56] | 42.2 | **63.4** | 44.7 | 26.5 | 50.9 | 59.1 | 30.6 | 56.1 | 30.0 | 26.3 | 46.0 | 33.1 |
| + BPR [47] | 42.8 | 63.3 | 45.3 | 26.1 | 51.0 | 60.6 | 32.0 | **57.3** | 31.6 | 25.9 | 46.3 | 36.3 |
| + **SegRefiner** (ours) | **43.6** | 63.3 | **45.2** | **27.4** | **51.4** | **61.6** | **34.1** | 57.2 | **34.9** | **27.2** | **47.1** | **39.9** |

Table 3: Tansfering our SegRefiner to other instance segmentation models.

| Method | Mask AP | | | | Boundary AP | | | |
|---|---|---|---|---|---|---|---|---|
| | AP | $AP_S$ | $AP_M$ | $AP_L$ | AP | $AP_S$ | $AP_M$ | $AP_L$ |
| PointRend [29] | 41.5 | 25.1 | 49.0 | 59.3 | 30.6 | 25.0 | 44.2 | 34.1 |
| + SegRefiner | 42.8 +1.3 | 25.9 +0.8 | 50.4 +1.4 | 61.3 +2.0 | 33.7 +3.1 | 25.8 +0.8 | 46.2 +2.0 | 40.1 +6.0 |
| RefineMask [57] | 41.2 | 24.0 | 48.1 | 59.2 | 30.5 | 23.8 | 43.5 | 34.1 |
| + SegRefiner | 41.9 +0.7 | 24.6 +0.6 | 48.7 +0.6 | 60.7 +1.5 | 33.0 +2.5 | 24.5 +0.7 | 44.7 +1.2 | 39.4 +5.3 |
| Mask Transfiner [27] | 42.2 | 25.9 | 49.0 | 60.1 | 31.6 | 25.8 | 44.5 | 35.8 |
| + SegRefiner | 43.3 +1.1 | 26.8 +0.9 | 49.9 +0.9 | 62.0 +1.9 | 34.4 +2.8 | 26.6 +0.8 | 45.9 +1.4 | 41.3 +5.5 |
| SOLO [51] | 37.4 | 19.3 | 45.5 | 56.6 | 24.7 | 19.0 | 39.3 | 27.8 |
| + SegRefiner | 40.5 +3.1 | 21.7 +2.4 | 49.3 +3.8 | 60.9 +4.3 | 31.3 +6.6 | 21.4 +2.4 | 44.8 +5.5 | 39.8 +12.0 |
| CondInst [48] | 39.8 | 24.3 | 47.5 | 54.8 | 29.2 | 24.1 | 42.6 | 30.2 |
| + SegRefiner | 41.1 +1.3 | 25.5 +1.2 | 48.8 +1.3 | 56.8 +2.0 | 32.2 +2.6 | 25.3 +1.2 | 44.5 +1.9 | 36.1 +5.9 |
| QueryInst [18] | 42.4 | 26.7 | 49.5 | 61.4 | 29.9 | 26.4 | 44.5 | 32.3 |
| + SegRefiner | 44.2 +1.8 | 27.5 +0.8 | 51.4 +1.9 | 64.7 +3.3 | 34.9 +5.0 | 27.3 +0.9 | 47.2 +2.7 | 42.6 +10.3 |
| Mask2Former [11] | 46.8 | 27.9 | 54.9 | 69.1 | 37.0 | 27.8 | 50.1 | 44.6 |
| + SegRefiner | 47.4 +0.6 | 28.5 +0.6 | 55.3 +0.4 | 69.8 +0.7 | 38.8 +1.8 | 28.4 +0.6 | 51.0 +0.9 | 48.5 +4.2 |

Table 4: Transfering our SegRefiner to DIS task. SegRefiner is employed to refine the segmentation results from 6 different models. w/o and w/ indicates the original and refined results, respectively.

| Dataset | Metrics | U-Net [42] | | PFNet [36] | | PSPNet [60] | | ICNet [59] | | HRNet[50] | | ISNet[44] | |
|---|---|---|---|---|---|---|---|---|---|---|---|---|---|
| | | w/o | w/ | w/o | w/ | w/o | w/ | w/o | w/ | w/o | w/ | w/o | w/ |
| DIS-VD | IoU | 54.77 | 58.73 +3.96 | 56.20 | 60.44 +4.24 | 56.41 | 60.41 +4.00 | 56.47 | 61.38 +4.91 | 61.02 | 64.43 +3.41 | 67.10 | 68.27 +1.17 |
| | mBA | 69.84 | 75.44 +5.6 | 63.69 | 74.83 +11.14 | 62.78 | 74.84 +12.06 | 66.47 | 75.55 +9.08 | 68.94 | 76.62 +7.68 | 74.13 | 76.64 +2.51 |
| DIS-TE1 | IoU | 44.11 | 47.05 +2.94 | 46.97 | 49.01 +2.04 | 47.57 | 49.01 +1.44 | 46.22 | 49.74 +3.52 | 50.98 | 52.97 +1.99 | 56.17 | 57.00 +0.83 |
| | mBA | 70.13 | 74.97 +4.84 | 65.13 | 74.67 +9.54 | 64.45 | 74.67 +10.22 | 67.19 | 75.01 +7.82 | 69.34 | 75.62 +6.28 | 73.63 | 75.49 +1.86 |
| DIS-TE2 | IoU | 54.39 | 58.10 +3.71 | 57.04 | 60.26 +3.22 | 57.71 | 60.25 +2.54 | 56.50 | 60.33 +3.83 | 61.22 | 64.70 +3.48 | 67.05 | 67.84 +0.79 |
| | mBA | 69.99 | 75.57 +5.58 | 64.64 | 75.05 +10.41 | 63.71 | 75.06 +11.35 | 66.79 | 75.57 +8.78 | 69.50 | 76.64 +7.14 | 74.03 | 76.35 +2.32 |
| DIS-TE3 | IoU | 59.29 | 63.34 +4.05 | 59.46 | 63.42 +3.96 | 59.90 | 63.44 +3.54 | 59.83 | 64.20 +4.37 | 64.43 | 67.57 +3.14 | 69.94 | 70.87 +0.93 |
| | mBA | 70.59 | 76.68 +6.09 | 64.00 | 76.06 +12.06 | 62.74 | 76.05 +13.31 | 66.73 | 76.55 +9.82 | 69.48 | 77.60 +8.12 | 74.65 | 77.39 +2.74 |
| DIS-TE4 | IoU | 61.14 | 66.29 +5.15 | 59.11 | 66.63 +7.52 | 57.61 | 66.64 +9.03 | 60.53 | 67.01 +6.48 | 64.67 | 70.47 +5.80 | 70.12 | 72.21 +2.09 |
| | mBA | 70.68 | 77.07 +6.39 | 62.78 | 76.43 +13.65 | 61.88 | 76.43 +14.55 | 66.42 | 77.06 +10.64 | 68.09 | 77.86 +9.77 | 74.35 | 77.66 +3.31 |

demands on a model's capability to perceive and capture intricate information. Thus, it serves as a suitable benchmark to evaluate the performance of our refinement method. The evaluation metrics and other settings utilized in this experiment are consistent with those in Sec. 4.2.

**Results** Since the recent introduction of DIS5K, no prior refinement methods have reported results on this dataset to date. Therefore, the main aim of this experiment is to evaluate the transferability of our SegRefiner across various models. As shown in Tab. 4, our SegRefiner is applied to 6 segmentation models. The results consistently demonstrate that our SegRefiner improves the performance of each segmentation model in terms of both IoU and mBA. In Fig. 4, we provide qualitative comparisons with previous methods, which reveal the superior ability of our SegRefiner to capture fine-grained details, such as the dense and fine mesh of the chair.

## 4.5 Ablation Study

We conduct ablation studies on the BIG dataset [44] with high-resolution images. We first investigate the effectiveness of the diffusion process in SegRefiner. The results are reported in Tab. 5a, where the term "none" refers to the utilization of a U-Net without multi-step iteration, and "w/o diffusion" denotes the direct utilization of the previous step's mask as the input for the subsequent iteration, without employing the diffusion sampling process, *i.e.*, discarding the transition sample module. Since there is no states-transition probability in non-diffusion methods, we use a sliding window strategy similar to [12] in this experiment to obtain the input for local refinement. The results in Tab. 5a demonstrate that the diffusion-based iterative process achieves the best performance, validating the effectiveness of the diffusion process in SegRefiner.

In Tab. 5b and Tab. 5c, we evaluate various alternatives used in prior experiments. Tab. 5b shows the analysis of global and local refinement used in high resolution images. As can be seen, global refinement significantly improves IoU; however, for high-resolution images, the resulting smaller output size leads to a lower mBA. Local refinement applied to local patches of the original size greatly improves mBA, while its enhancement in IoU is less significant due to the absence of global information. The combination of local and global refinement achieves better performance in both IoU and mBA. Tab. 5c presents the results corresponding to different input image sizes. Considering computational load and memory usage, $256 \times 256$ is selected as the default setting, which performs well without introducing too much computational overhead.

Table 5: Ablation studies on BIG dataset [44]. The best results are highlighted in **bold** and the default settings are marked with gray.

(a) Effectiveness of the diffusion process.

| Iteration Type | IoU | mBA |
|---|---|---|
| none | 94.10 | 75.03 |
| w/o diffusion | 94.11 | 75.10 |
| w/ diffusion | **94.85** | **77.64** |

(b) Ablation on global and local refinement.

| Method | IoU | mBA |
|---|---|---|
| only global | 92.43 | 60.58 |
| only local | 89.73 | 63.56 |
| global+local | **94.85** | **77.64** |

(c) Ablation on input size.

| Input Size | IoU | mBA |
|---|---|---|
| 128×128 | 93.10 | 74.28 |
| 256×256 | 94.85 | **77.64** |
| 512×512 | **94.97** | 77.56 |

# 5 Conclusion and Discussion

We propose SegRefiner, which is the first diffusion-based image segmentation refinement method with a new designed discrete diffusion process. SegRefiner performs model-agnostic segmentation refinement and achieves strong empirical results in refinement of various segmentation tasks.

While SegRefiner has achieved significant improvements in accuracy, **one limitation** lies in that the diffusion process leads to slowdown of the inference due to the multi-step iterative strategy. As shown in Tab. 6, we conduct an experiment on instance segmentation about the relationship between the model's accuracy, computational complexity, time consumption (the average time consumed per image), and the number of iteration steps. It can be observed that, while the iterative strategy has provided SegRefiner with a noticeable improvement in accuracy, it has also introduced a linear increase with the number of steps in both time consumption and computational complexity. As the first work applying diffusion models to the refinement task, the proposed SegRefiner primarily concentrates on devising a suitable diffusion process for general refinement tasks. While improving the efficiency of diffusion models will be a crucial research direction in the future, not only in the field of image generation but also in other domains where diffusion models are applied.

Table 6: The relationship between SegRefiner's accuracy, efficiency and the number of iteration steps on instance segmentation.

| Steps | 0 | 1 | 2 | 3 | 4 | 5 | 6 |
|---|---|---|---|---|---|---|---|
| Mask AP | 39.8 | 40.7 | 41.0 | 41.4 | 41.6 | 41.9 | 41.9 |
| Boundary AP | 27.3 | 29.9 | 30.6 | 31.5 | 32.1 | 32.5 | 32.6 |
| Time (s) | n/a | 0.52 | 1.01 | 1.53 | 1.98 | 2.45 | 2.94 |
| GFLOPs | n/a | 249 | 497 | 746 | 994 | 1243 | 1492 |

# Acknowledgement

This research was funded by the Fundamental Research Funds for the Central Universities (2023JBZD003).

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

# Appendix

Here we first introduce the details of SegRefiner's equation derivation in Sec. A. The implementation details are shown in Sec. B. Moreover, we provide additional qualitative results in Sec. C.

## A  Equation Derivation Details

In the following, we will present the derivation of the posterior distribution (Eq. (9)) and the reverse transition-probability (Eq. (11)). Firstly, for Eq. (9), it's derived from the Bayes' theorem:

$$q(x_{t-1}|x_t, x_0) = \frac{q(x_t|x_{t-1}, x_0)q(x_{t-1}|x_0)}{q(x_t|x_0)}. \tag{12}$$

According to Eq. (6) and Eq. (7), we can get all the items in Bayes' theorem as:

$$q(x_t|x_{t-1}, x_0) = x_{t-1}Q_t x_t^T, \tag{13}$$

$$q(x_{t-1}|x_0)) = x_0\bar{Q}_{t-1}x_{t-1}^T, \tag{14}$$

$$q(x_t|x_0)) = x_0\bar{Q}_t x_t^T. \tag{15}$$

It appears to be slightly different from Eq. (6). The reason is that the results of Eq. (6) is a length-2 vector, representing the probabilities of two states respectively. While we use a scalar form here, which includes an additional one-hot vector $x_*^T$ and is more suitable for subsequent derivations. For the denominator of Eq. (9), we have obtained the same form. Let's consider the numerator:

$$\begin{aligned}
q(x_t|x_{t-1}, x_0)q(x_{t-1}|x_0) &= x_{t-1}Q_t x_t^T \cdot x_0\bar{Q}_{t-1}x_{t-1}^T \\
&= x_{t-1}(x_t Q_t^T)^T \cdot x_0\bar{Q}_{t-1}x_{t-1}^T \\
&= (x_t Q_t^T)x_{t-1}^T \cdot (x_0\bar{Q}_{t-1})x_{t-1}^T \\
&= [(x_t Q_t^T) \odot (x_0\bar{Q}_{t-1})]x_{t-1}^T,
\end{aligned} \tag{16}$$

where "·" refers to scalar multiplication and "⊙" refers to element-wise multiplication. This is the scalar form, by removing the $x_{t-1}^T$, we can get the corresponding vector form just as Eq. (9).

Secondly, for Eq. (11), it's derived from Eq. (9) by substituting $x_0$ with $[p_\theta(\tilde{m}_{0|t})^{i,j}, 1 - p_\theta(\tilde{m}_{0|t})^{i,j}]$ (we refer $p_\theta(\tilde{m}_{0|t})^{i,j}$ to $p_0$ in the following for simplicity).

$$\begin{aligned}
p(x_{t-1}|x_t) &= \frac{[(x_t Q_t^T) \odot (x_0\bar{Q}_{t-1})]x_{t-1}^T}{x_0\bar{Q}_t x_t^T}, \ \ with \ x_0 = [p_0, 1-p_0] \\
&= \frac{x_t(Q_t^T \odot [p_0\bar{\beta}_{t-1}, 1 - p_0\bar{\beta}_{t-1}])x_{t-1}^T}{[p_0\bar{\beta}_t, 1 - p_0\bar{\beta}_t]x_t^T} \\
&= \frac{x_t \begin{bmatrix} p_0\bar{\beta}_t & 0 \\ p_0(\bar{\beta}_{t-1} - \bar{\beta}_t) & 1 - p_0\bar{\beta}_{t-1} \end{bmatrix} x_{t-1}^T}{x_t[p_0\bar{\beta}_t, 1 - p_0\bar{\beta}_t]^T} \\
&= x_t \begin{bmatrix} 1 & 0 \\ \frac{p_0(\bar{\beta}_{t-1}-\bar{\beta}_t)}{1-p_0\bar{\beta}_t} & \frac{1-p_0\bar{\beta}_{t-1}}{1-p_0\bar{\beta}_t} \end{bmatrix} x_{t-1}^T.
\end{aligned} \tag{17}$$

Same as Eq. (9), we can remove $x_{t-1}^T$ and get the vector form of Eq. (11).

## B  Implementation Details

In this section, we give a detailed description of the model architecture and training/inference settings. The overall workflow of the training and inference process are provided in Alg. 1 and Alg. 2.

**Model Architecture**   Following [39], we use a U-Net with 4-channel input and 1-channel output. Both input and output resolution is set to $256 \times 256$. Considering computational load and memory usage, we set the intermediate feature channels to 128 and only conduct *self-attention* in strides 16 and 32.

**Algorithm 1** Training

**Input** total diffusion steps $T$, datasets $D = \{(I, M_{fine}, M_{coarse})^K\}$
**repeat**
    Sample $(I, M_{fine}, M_{coarse}) \sim D$
    Sample $t \sim Uniform(1, \ldots, T)$
    Initialize $m_0 = M_{fine}, x_0^{i,j} = [1, 0]$
    $q(x_t^{i,j}|x_0^{i,j}) = x_0^{i,j}\bar{Q}_t$
    Sample $x_t^{i,j} \sim q(x_t^{i,j}|x_0^{i,j})$, get $x_t \in \{0,1\}^{2 \times H \times W}$
    Pixels Transition $m_t = x_t[0] \odot M_{fine} + x_t[1] \odot M_{coarse}$
    Take gradient descent step on $\nabla_\theta \mathcal{L}(f_\theta(I, m_t, t), M_{fine})$
**until** convergence

---

**Algorithm 2** Inference

**Input** total diffusion steps $T$, image and coarse mask $(I, M_{coarse})$
**Initialize** $x_T = [0, 1], m_T = M_{coarse}$
**for** $t$ in $\{T, T-1, \ldots, 1\}$ **do**
    $\tilde{m}_{0|t}, \; p_\theta(\tilde{m}_{0|t}) = f_\theta(I, m_t, t)$
    $p_\theta(x_{t-1}^{i,j}|x_t^{i,j}) = x_t^{i,j} P_{\theta,t}^{i,j}$
    Sample $x_t^{i,j} \sim p_\theta(x_{t-1}^{i,j}|x_t^{i,j})$, get $x_t \in \{0,1\}^{2 \times H \times W}$
    Pixels Transition $m_{t-1} = x_{t-1}[0] \odot \tilde{m}_{0|t} + x_{t-1}[1] \odot M_{coarse}$
**return** $m_0$

**Training Settings** All experiments are conducted on 8 NVIDIA RTX3090 GPUs with Pytorch. During training, we first train the *LR-SegRefiner* on the LVIS dataset [20] with 120k iterations. The AdamW optimizer is used with the initial learning rate of $4 \times 10^{-4}$. We use a multi-step learning rate schedule, which decays by 0.5 in steps 80k and 100k. Subsequently, the *HR-SegRefiner* is obtained from 40k-iterations fine-tuning based on the 80k checkpoint of *LR-SegRefiner*. Batch size is set to 8 in each GPU.

**Inference Settings** In instance segmentation, we use the *LR-SegRefiner* to perform refinement in instance level. For each instance, we extract the bounding box region based on the coarse mask and expand it by 20 pixels on each side. The extracted region is then resized to match the input size of the model. After a complete reverse diffusion process, the output is resized to the original size.

In semantic segmentation and dichotomous image segmentation, because of the high resolution of images, we employ the *HR-SegRefiner* and conduct a global-and-local refinement process. In order to identify the local patches that require refinement, we filter out pixels with low state-transition probabilities from the globally refined mask and use them as the center points for the local patches. We apply Non-Maximum Suppression (NMS, with 0.3 as threshold) to these patches to remove excessive overlapping.

## C Qualitative Results

In this section, we provide more visual results in semantic segmentation, instance segmentation, and dichotomous image segmentation. Fig. 5 shows the comparisons of SegRefiner and other models (including instance segmentation models and refinement models) on COCO [34] validation set. Fig. 6 shows more comparisons between the coarse masks and refined masks on COCO validation set. These results demonstrate that the proposed SegRfiner can robustly correct inaccurate predictions in coarse masks. Fig. 7 and Fig. 8 show visual results on BIG dataset [12] and DIS5K dataset [40]. SegRefiner shows a strong capability for capturing extremely fine details on these two high-resolution datasets.

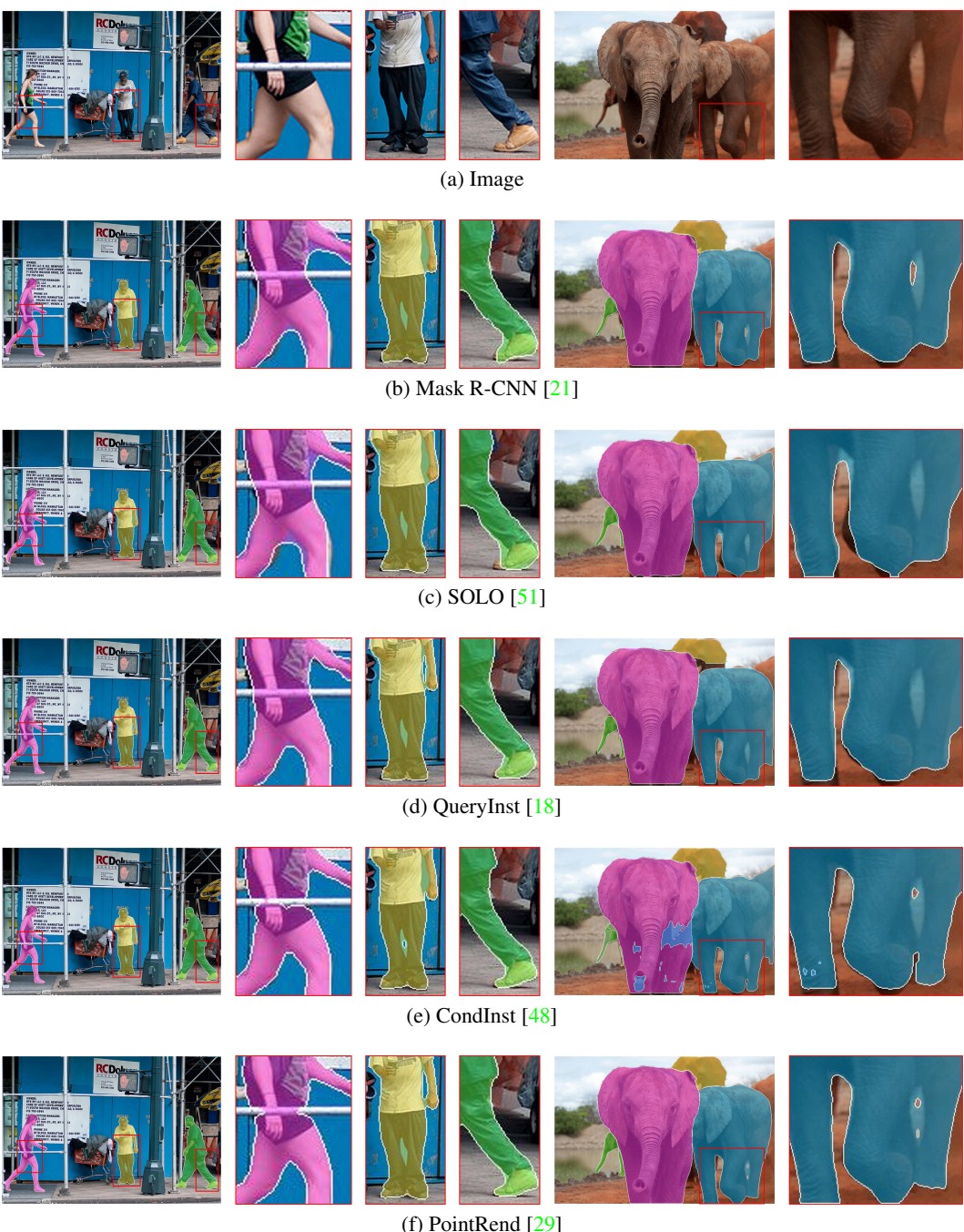

(a) Image

(b) Mask R-CNN [21]

(c) SOLO [51]

(d) QueryInst [18]

(e) CondInst [48]

(f) PointRend [29]

Figure 5: Visual comparisons with other instance segmentation and refinement methods on COCO dataset. Our SegRefiner can robustly correct prediction errors both outside and inside the coarse mask. (Please refer to the next page for the remaining portion of this figure.)

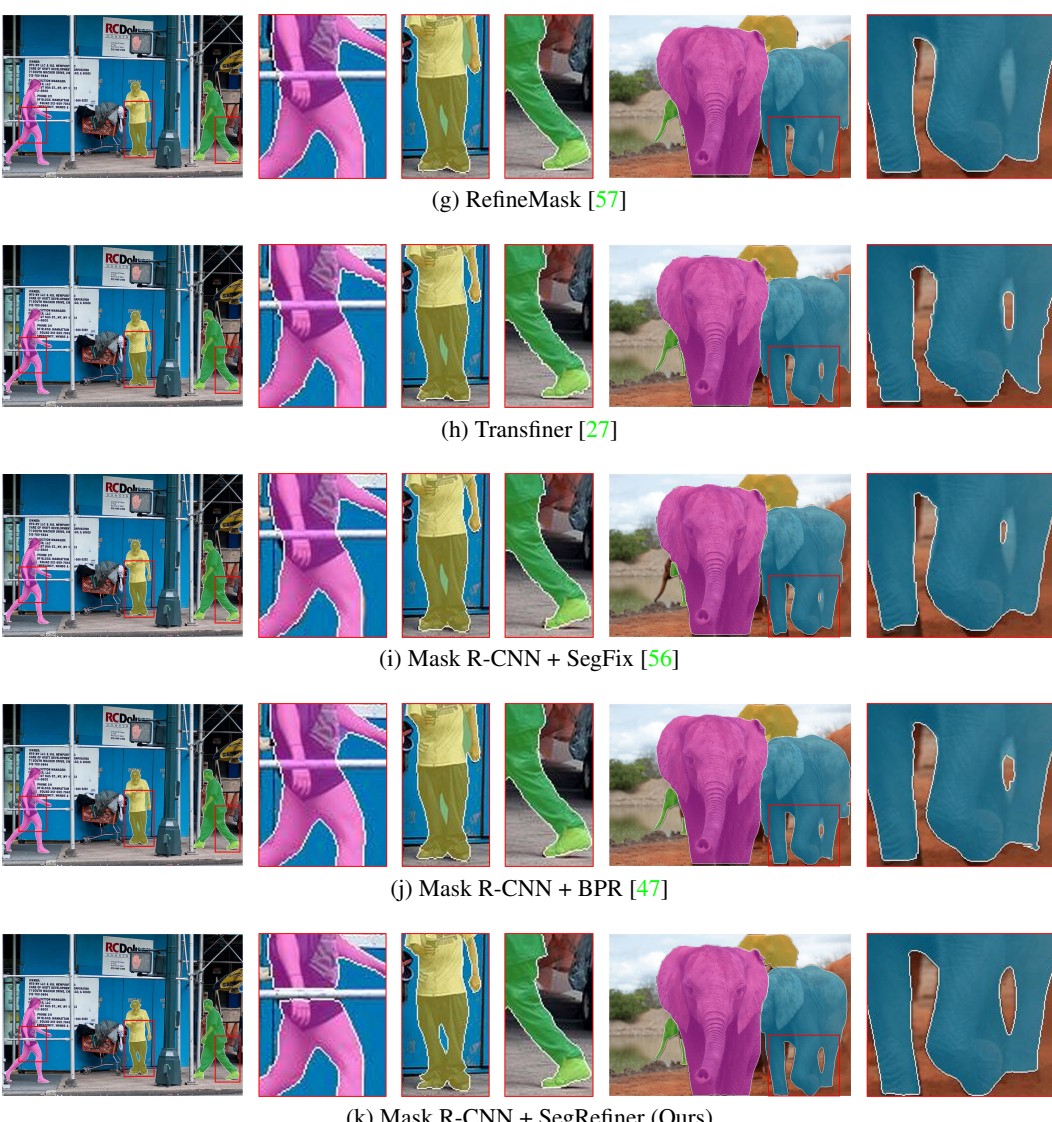

Figure 5: Visual comparisons with other instance segmentation and refinement methods on COCO dataset. Our SegRefiner can robustly correct prediction errors both outside and inside the coarse mask.

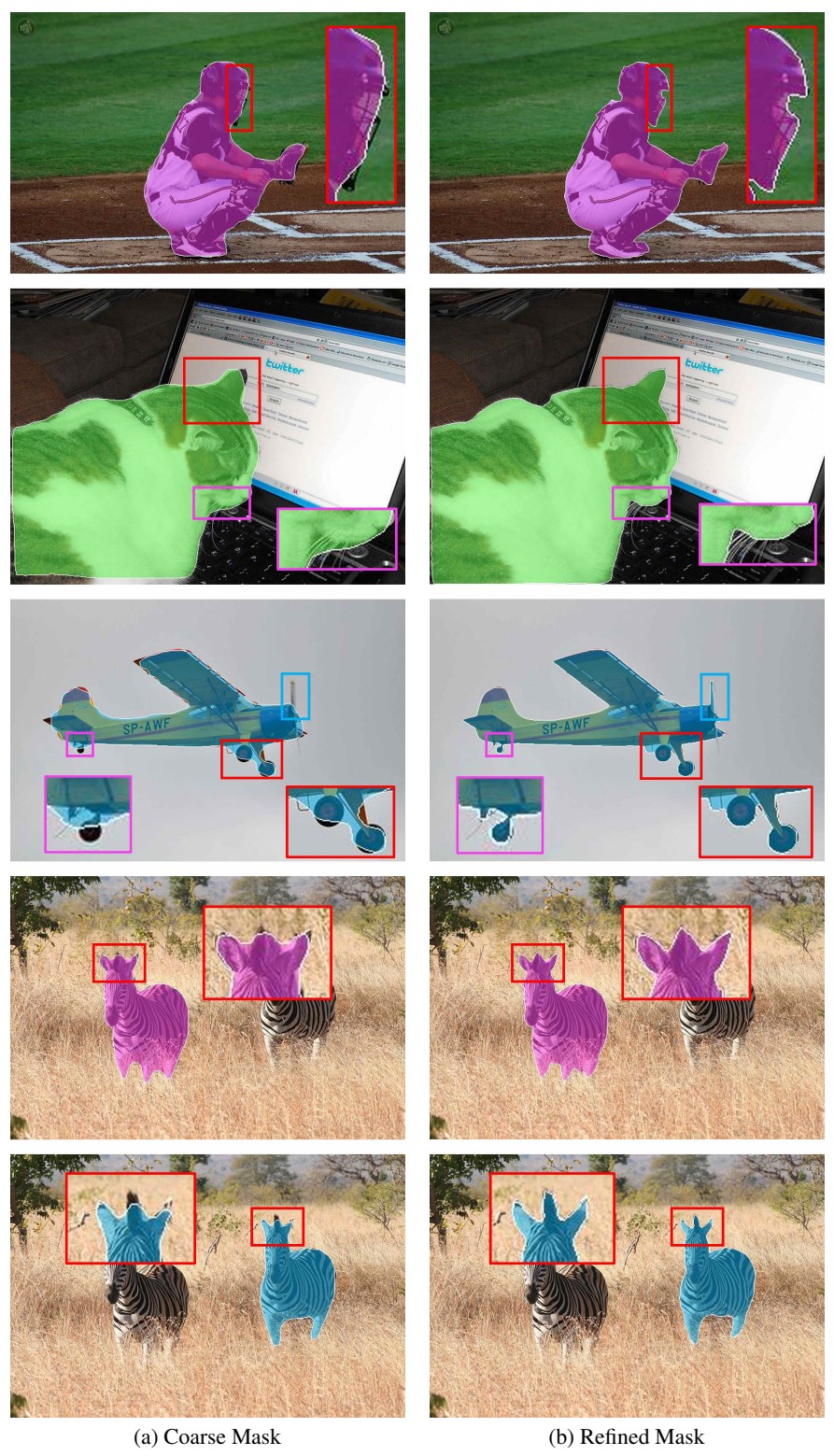

(a) Coarse Mask                    (b) Refined Mask

Figure 6: More visual results on COCO dataset. Coarse masks are obtained from Mask R-CNN [21]. Our SegRefiner corrects the errors of coarse masks (see Refined Mask).

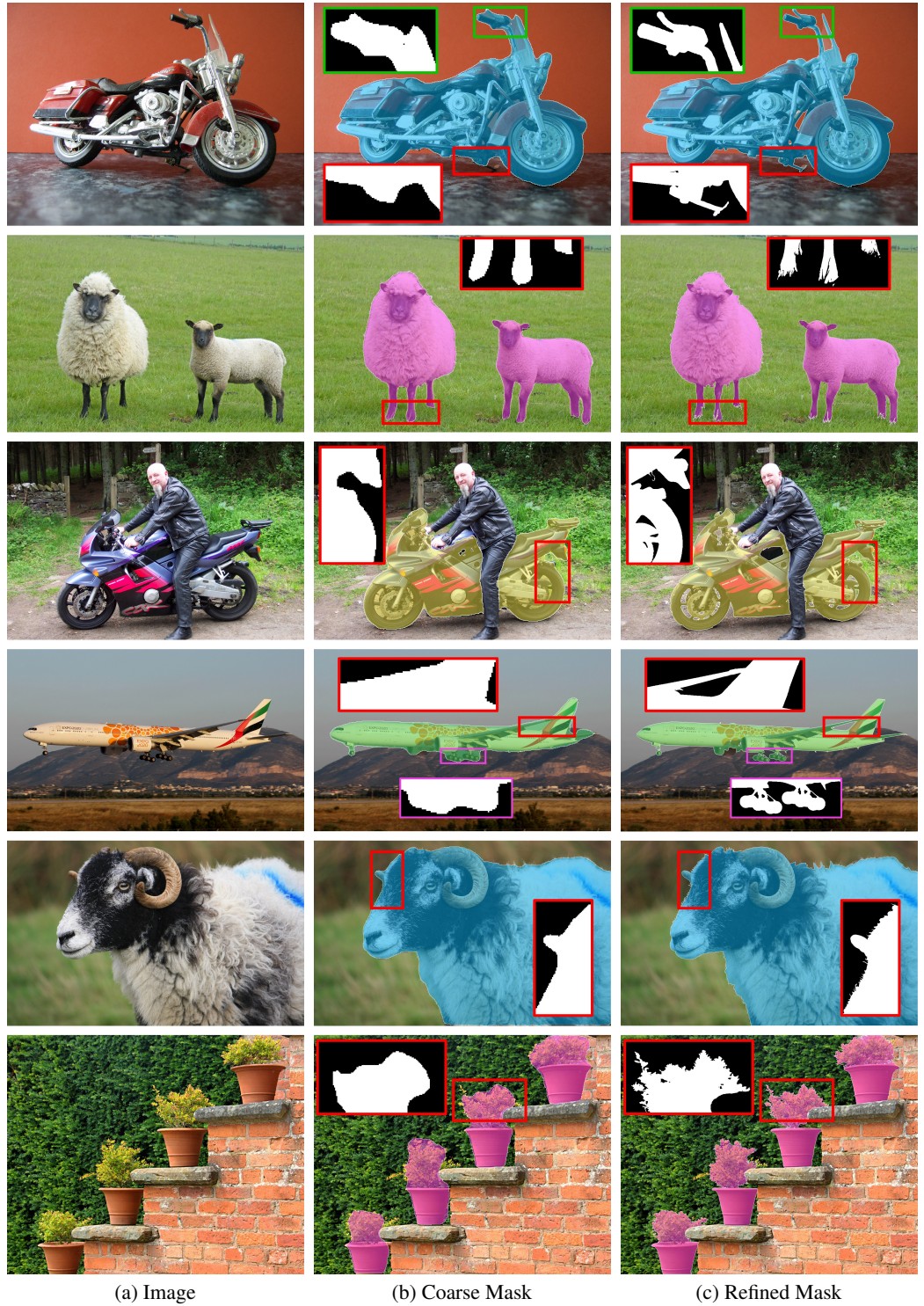

(a) Image       (b) Coarse Mask       (c) Refined Mask

Figure 7: More visual results on BIG dataset [12]. Coarse masks are obtained from Deeplab v3+ [5]. Our SegRefiner greatly enhances the mask quality (see Refined Mask). Please kindly zoom in for a better view.

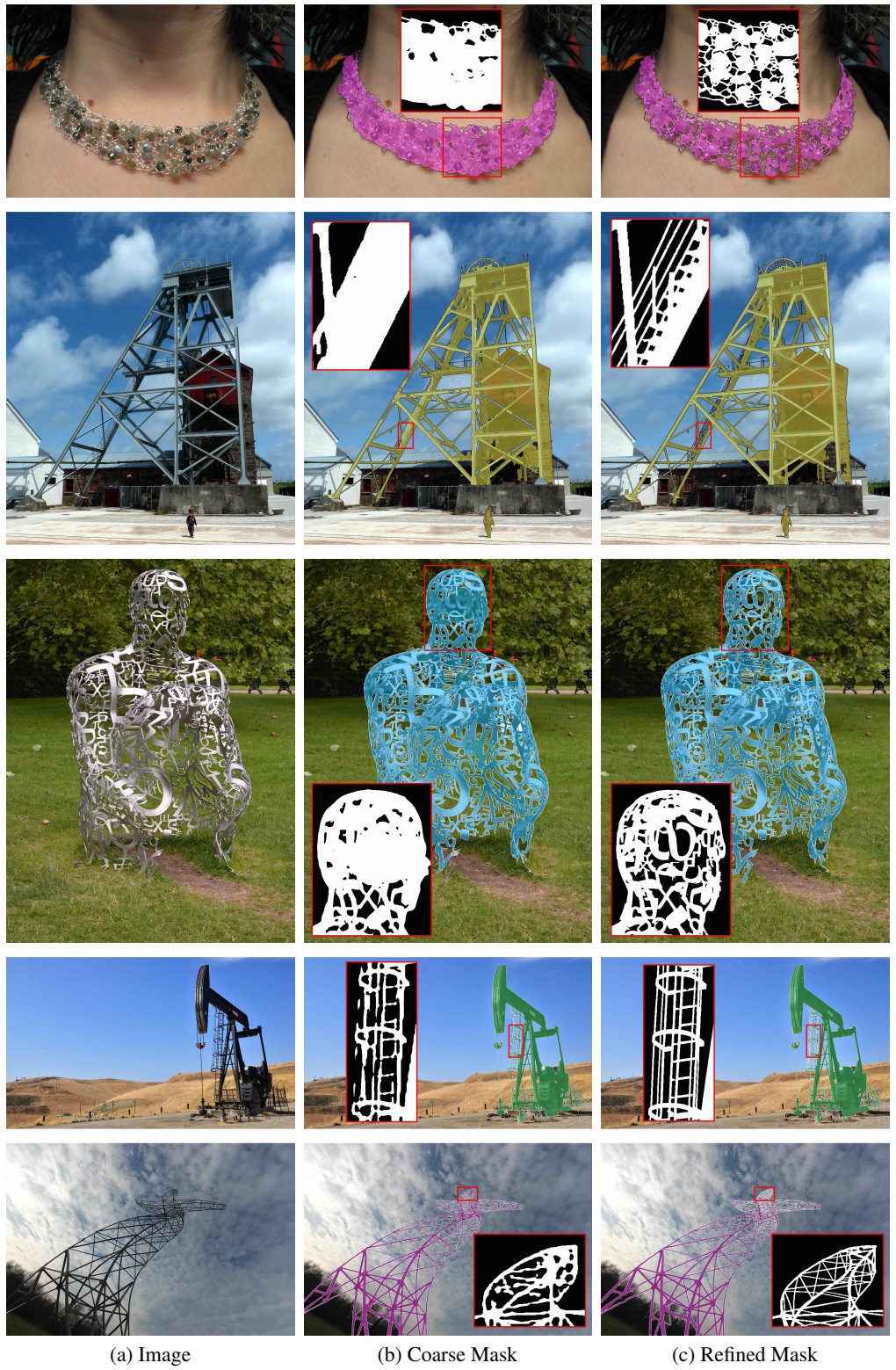

| (a) Image | (b) Coarse Mask | (c) Refined Mask |

Figure 8: More visual results on DIS5K dataset [40]. Coarse masks are obtained from ISNet [40]. Our SegRefiner captures extremely fine details (see Refined Mask). Please kindly zoom in for a better view.

