# OpenReview forum: "SegRefiner: Towards Model-Agnostic Segmentation Refinement with Discrete Diffusion Process"
_NeurIPS.cc/2023/Conference — NeurIPS 2023 poster_

### Official Review · Reviewer_Xhma · 2023-07-04

**Soundness:** 3 good
**Presentation:** 3 good
**Contribution:** 3 good
**Rating:** 6
**Confidence:** 3

**Summary:**

This paper develops a method based discrete diffusion process to refine coarse segmentation masks with input images in the context of semantic image segmentation. In this work, the refinement problem has been formulated as an iterative data generation process. To work with binary masks, as adapted for most segmentation scenarios, the authors proposed discrete state translation with mathematical analysis. Experimental results seem solid about the superiority of the proposed methods against several parallel methods on three segmentation tasks.

**Strengths:**

Originality: this paper is new for its adaptation of discrete prior to tackling binary segmentation masks. although the discrete diffusion model is not new, its application in segmentation can be an interesting attempt.  In order to handle a smooth translation from a fine mask to a coarse one, the authors proposed a transition sample module in the forward process. This is different from adding traditional Gaussian noise as in continuous diffusion models.

Quality: the quality of this paper is relatively good, through multiple reading passes. The paper is kind of self-contained, including strong motivation, mathematical analysis, and experimental results.

Clarity: This paper is not easy to read for my first pass. After reading it several times, one can have a better understanding of certain concepts and illustrations

Significance: I give high significance to this paper as it solves a common problem in a novel way as well as the obtained results seem to be greatly improved in a wide range of segmentation tasks and datasets.

**Weaknesses:**

Personally, the biggest concern of this paper is the clarification. Basically, the authors devised a discrete process to bridge the fine mask with the coarse one, by using a transitioned mask. This mask is unidirectional, which means only the transition to the target happens.  This forward process consists of a continuous Q_{t} and a discrete sampling based on probability defined on Q_{t}. These processes are not well illustrated and explained in the paper. In addition, the paper refers to source and target masks differently for forward and backward processes, which can confuse readers and a better nomination would be suggested.

Although this paper presents extensive experimental results, it lacks a comparison of computational complexity and time and resources. Despite the fact that the authors had a small discussion on this aspect in the conclusion, it is much better to provide quantitative results either in the main paper or in the supplementary material.

**Questions:**

- throughout the paper, the authors utilized examples with binary masks to illustrate the idea, including Fig. 2, Equ. 6-11, etc. Could the authors explain how exactly to apply this method to a multi-class segmentation problem?

- Since the authors utilize Gumbel-softmax sampling in the forward, it might exhibit some kind of diversities for the transitioned masks in the intermediate steps (even the final output–coarse mask remains the same). However, for the backward process, if my understanding is not wrong, how can this algorithm ensure a certain degree of diversity as no stochastic operations are involved?

- The forward process has ground truth for the source and target masks, while for the backward process, the paper applied simple thresholding. This can cause instability of convergence. Did the authors meet any optimization difficulties?

- In the training strategy, why exactly two versions of SegRefiner were proposed?

**Limitations:**

The authors have adequately addressed the limitation of this paper while I am expecting more details, as stated in the previous comments.

---

> ### Author Rebuttal · Authors · 2023-08-10
>
> # Response to Reviewer Xhma
> ### **Q1. Clarification of Method Description**
> Thank you for your valuable and thoughtful comments and we will rewrite the Method section to enhance clarification. Due to the confusion caused by "source and target masks", we will replace them with "fine and coarse masks" in both forward and reverse process. The description of the overall method will be modified as follows:
> * In the forward process, fine mask corresponds to the ground truth. The forward process involves binary sampling based on the probability $Q_t$ (Eq. 6), gradually transforming the fine mask into a coarse one, yielding masks of varying coarseness levels used for training.
> * In the reverse process, coarse mask is the previous segmentation result which needed to be refined, while the fine mask comes from our model's prediction. The reverse process involves sampling according to the probability $P_t$ (Eq. 11) and progressively correcting coarse predictions.
> ### **Q2. Quantitative results of accuracy, computational complexity and time**
> Table below shows the relationship between the model's accuracy, efficiency and the number of iteration steps on instance segmentation. It can be observed that both time consumption and computational complexity increase linearly with the number of steps, while accuracy reaches saturation around 5-6 steps.
>
> | Steps       |  0   |  1   |  2   |  3   |  4   |  5   |  6   |  7   |  8   |
> |:----------- |:----:|:----:|:----:|:----:|:----:|:----:|:----:|:----:|:----:|
> | Mask AP     | 39.8 | 40.7 | 41.0 | 41.4 | 41.6 | 41.9 | 41.9 | 41.9 | 41.9 |
> | Boundary AP | 27.3 | 29.9 | 30.6 | 31.5 | 32.1 | 32.5 | 32.6 | 32.6 | 32.6 |
> | Time(s)     | n/a  | 0.52 | 1.01 | 1.53 | 1.98 | 2.45 | 2.94 | 3.44 | 3.92 |
> | FLOPs(G)    | n/a  | 249  | 497  | 746  | 994  | 1243 | 1492 | 1740 | 1989 |
>
> We also provide a comparison between our method and previous methods in the following table. For semantic segmentation:
>
> | Method     | Time(s) | FLOPs(G) | Params(M) |
> |:---------- |:-------:|:--------:|:---------:|
> | CascadePSP |   6.2   |  26518   |   67.62   |
> | CRM        |   4.3   |   2356   |   9.27    |
> | **Ours**   |   1.6   |   1492   |  118.93   |
>
> For instance segmentation:
>
> | Method   | Time(s) | FLOPs(G) | Params(M) |
> |:-------- |:-------:|:--------:|:---------:|
> | BPR      |   1.4   |   135    |   65.85   |
> | **Ours** |   2.9   |   1492   |  118.93   |
>
> Results indicate that, in semantic segmentation, our method demonstrates faster speed and lower computational complexity. However, in instance segmentation, our method is slower than previous approaches.
> This is largely due to the high computational complexity of the U-Net model used in previous diffusion-based works (with 249 GFLOPs per step). While this work focuses on exploring the feasibility and superiority of diffusion process in refinement task, thus we simply maintain the model structure unchanged. In the future, we will continue to explore more efficient segmentation refinement by replacing the U-Net with a more lightweight architecture.
> ### **Q3. Applying to multi-class segmentation problem**
> As mentioned in the paper, our method refines coarse segmentation results in a class-agnostic manner. Its mechanism is first obtaining prompts from the coarse mask to determine the objects needed to be segmented and then providing accurate masks. Thus our method only needs to perform binary foreground-background prediction without the need to predict categories.  As for coarse segmentation results with multiple classes, the proposed method can refine the coarse mask of every class sequentially.
> ### **Q4. Gumbel-softmax. Stochastic operations in backward process**
> In the backward process (referred as reverse process in the paper), there are also stochastic operations. During the backward process, our method first get the predicted fine mask and the reversed states-transition probability as shown in Eq.(8)-(11). Then, it performs Gumbel-softmax sampling on this probability and replace the sampled pixels with the values of the predicted fine mask. This process is also stochastic, ensuring a certain degree of diversity.
> ### **Q5. Thresholding in backward process**
> We are sorry that we may not have fully understood your meaning. If the "backward process" you referred to is the reverse diffusion process, it's the inference process of our method and does not affect the stability of convergence and optimization of training.
> While there is another similar concern. As you mentioned, during the inference process, we replace the ground truth with a predicted mask, which could potentially introduce a gap between training and inference. To address this issue, we propose the states-transition strategy shown in Eq. (11). The states-transition probability is related to the confidence of the prediction, ensuring that pixels with high confidence are prioritized to be replaced by the predicted values. This helps to bridge the gap between training and inference.
> ### **Q6. Why two versions**
> This is mainly because low-resolution and high-resolution segmentation tasks have different characteristics, resulting in significant distribution gap between corresponding datasets.
> * Low-resolution segmentation focus on accurately identifying foreground objects without introducing semantic errors, and the datasets exhibit complex scenes with multiple objects.
> * High-resolution segmentation emphasizes capturing very fine details (such as hair, railings, etc.), and the datasets are object-centric.
>
> If mixing these two types of datasets together for training, the distribution gap can lead to optimization difficulties and affecting the model's accuracy on both low and high resolutions. Therefore, in this work, we have chosen the simplest approach: separating it into two versions. While it's possible to combine different-resolution datasets for training with a more carefully crafted strategy, and we will continue to explore this issue in our future work.

---

> > ### Comment · Reviewer_Xhma · 2023-08-21
> > **Thank you for the answers**
> >
> > I thank the authors for their careful answers. I will increase my score to weakly accept.

---

> ### Comment · Area_Chair_geEb · 2023-08-18
> **Please acknowledge reading the rebuttal**
>
> Dear reviewer,
>
> Please acknowledge reading the rebuttal. One can acknowledge reading the rebuttal by posting an official comment on the open review platform.
>
> Best,
> AC

---

### Official Review · Reviewer_oFWK · 2023-07-09

**Soundness:** 3 good
**Presentation:** 3 good
**Contribution:** 3 good
**Rating:** 6
**Confidence:** 4

**Summary:**

In this paper, the authors propose a model-agnostic solution called SegRefiner to enhance the quality of object masks generated by different segmentation models. The approach introduces a novel perspective on segmentation refinement, treating it as a data generation process. By leveraging a series of denoising diffusion steps, SegRefiner achieves effective mask refinement.
Specifically, SegRefiner takes coarse masks as inputs and employs a discrete diffusion process to refine them. By predicting the label and states-transition probabilities for each pixel, the method progressively refines the noisy masks in a conditional denoising manner. To evaluate the performance of SegRefiner, extensive experiments are conducted on various segmentation tasks, including semantic segmentation, instance segmentation, and dichotomous image segmentation.
The experimental results demonstrate the superiority of SegRefiner in multiple aspects. Firstly, it consistently improves both the segmentation metrics and boundary metrics for different types of coarse masks. Secondly, it outperforms previous model-agnostic refinement methods by a significant margin. Lastly, SegRefiner exhibits a strong capability to capture extremely fine details when refining high-resolution images.

**Strengths:**

(1) This paper is the first one to introduce diffusion-based refinement for segmentation masks.
(2) The proposed method, called SegRefiner, is model-agnostic, and thus applicable across different segmentation models and tasks.
(3) The authors extensively analyze the performance of SegRefiner across various segmentation tasks, demonstrating that our SegRefiner not only outperforms all previous model agnostic refinement methods, but can also be effortlessly transferred to other segmentation tasks without any modification.

**Weaknesses:**

(1) In the Objective Function of sec 4.1, how do you specify the value of alpha? Do you have any ablation studies on the alpha?
(2) In Table 3, it seems that the improvement on Boundary AP is larger than Mask AP. Why is that?
(3) In Table 4, it seems that the improvement on ISNet is smaller than the other datasets. Could you explain the reason?
(4) Why do you only do the ablation studies on semantic segmentation task? Have you ever tried the ablation studies on instance segmentation and Dichotomous Image Segmentation? Have you ever tried ablation studies on the number of iterations and output size? For Table 5 (c), have you ever tried a larger input size, like 1024x1024?

**Questions:**

I'm positive about this paper. I really like the idea to apply diffusion model to segmentation task. However, I have some concerns in the Weaknesses. I hope to see the response from the authors for those questions. Thank you.

**Limitations:**

The diffusion process leads to slowdown of the inference due to the multi-step iterative strategy.

---

> ### Author Rebuttal · Authors · 2023-08-10
>
> # Response to Reviewer oFWK
> ### **Q1. Value of $\alpha$ in the Objective Function**
> Regarding your concern, we do the following ablation study on instance segmentation with coarse masks from MaskRCNN (ResNet-50). The first row of the Table below refers to not using texture loss for training, and the remaining three rows show results with different values of $\alpha$.
>
> | $\mathcal{L}_{texture}$ | Mask AP | Boundary AP |
> | ----------------------- |:-------:|:-----------:|
> | None                    |  41.5   |    31.5     |
> | $\alpha$ = 1            |  41.9   |    32.1     |
> | $\alpha$ = 5            |  **41.9**   |    **32.6**     |
> | $\alpha$ = 10           |  41.7   |    32.5     |
>
> It can be observed that, on one hand, texture loss $\mathcal{L}_{texture}$ effectively enhances the model's ability to capture boundary details. On the other hand, when the weight of texture loss becomes too large, it may compromise the model's accuracy.
>
> This is in line with intuition. Since texture loss only incurs loss at the edges of the segmentation results, it can enhance the model's sensitivity to capture edge information. However, as these edge details often stem from low-level features, an excessively large weight may cause the model to overly focus on low-level features and overlook semantic consistency, resulting in erroneous predictions.
>
> Therefore, in our experiments, we chose $\alpha=5$, which well balances the two losses.
>
> ### **Q2. Why Boundary AP improves more**
> We greatly appreciate your observation! The more enhancements on Boundary AP effectively showcase the SegRefiner's capacity to capture more edge details and better refine coarse predictions. As illustrated in Fig. 1 and 4 of the main paper, existing segmentation models can provide segmentation results that roughly resemble the real shape of objects, but exhibit significant coarseness in edge details. While Mask IoU/AP is not sensitive to these edge errors, as the pixels near edges constitute only a small portion of the foreground. Instead, Boundary IoU/AP is solely related to the prediction accuracy of the edge region, which better reflects the precision of segmentation results in terms of fine details. For a clear illustration of this, please refer to **Fig. 2 in the rebuttal PDF file of General Response**.
>
> ### **Q3. Why results on ISNet improve less**
> Since ISNet is specifically designed for the DIS task and utilizes a specialized method of intermediate supervision strategy, it is better suited for the DIS task compared to other baselines in Tab. 4. As a result, predictions of ISNet has fewer potential errors that need to be corrected, leading to smaller improvement with our method.
>
> ### **Q4. (a) Ablation studies on other tasks**
> There are two main reasons:
> * Compared to instance segmentation, semantic segmentation and Dichotomous Image Segmentation involve both global and local refinements, allowing for analyzing more variables in ablation studies.
> * Refinements on semantic segmentation and Dichotomous Image Segmentation share exactly the same experimental settings. While the dataset for semantic segmentation is smaller. Considering the time and computational resources required for experiments, conducting ablation studies on semantic segmentation can be more efficient.
>
> We have also conducted the ablation study on “effectiveness of the diffusion process” on instance segmentation to address your concern. As shown in following table, the results are consistent with those in Tab. 5(a) in the paper.
>
> | Iteration Type | Mask AP | Boundary AP |
> |:--------------:|:-------:|:-----------:|
> |      none      |  40.9   |    30.1     |
> | w/o diffusion  |  41.4   |    31.0     |
> |  w/ diffusion  |  41.9   |    32.6     |
>
>
> ### **Q4. (b) Ablation studies on iterations**
> We conduct the following experiment in the table below about the number of iteration steps. It can be observed that both time consumption and computational complexity increase linearly with the number of steps, while accuracy reaches saturation around 5-6 steps. Considering the time and computational resource consumption, we aim to keep the number of iterations as low as possible. Therefore, we set it to 6.
>
> | Steps       |  0   |  1   |  2   |  3   |  4   |  5   |  6   |  7   |  8   |
> |:----------- |:----:|:----:|:----:|:----:|:----:|:----:|:----:|:----:|:----:|
> | Mask AP     | 39.8 | 40.7 | 41.0 | 41.4 | 41.6 | 41.9 | 41.9 | 41.9 | 41.9 |
> | Boundary AP | 27.3 | 29.9 | 30.6 | 31.5 | 32.1 | 32.5 | 32.6 | 32.6 | 32.6 |
> | Time(s)     | n/a  | 0.52 | 1.01 | 1.53 | 1.98 | 2.45 | 2.94 | 3.44 | 3.92 |
> | FLOPs(G)    | n/a  | 249  | 497  | 746  | 994  | 1243 | 1492 | 1740 | 1989 |
>
>
> ### **Q4. \(c\) Ablation studies on output size**
> In Tab. 5\(c\), we conducted ablation studies only on three output sizes: 128, 256, and 512. The reason is that smaller sizes (e.g., 64) would result in less detailed output even on low-resolution datasets like COCO and LVIS, leading to the loss of many fine-grained details. On the other hand, larger sizes (e.g., 1024) are wasteful since they already exceed the original size of images in low-resolution datasets. Additionally, they would introduce unacceptable computational complexity and memory usage.

---

> ### Comment · Area_Chair_geEb · 2023-08-18
> **Please acknowledge reading the rebuttal**
>
> Dear reviewer oFWK,
>
> Please acknowledge reading the rebuttal. One can acknowledge reading the rebuttal by posting an official comment on the open review platform.
>
> Best,
> AC

---

> > ### Comment · Reviewer_oFWK · 2023-08-21
> > **Keep my rating**
> >
> > Thanks for the author's rebuttal. It has resolved all of my concerns and I don't have any other questions. I'll keep my rating for weak accept. Thank you.

---

### Official Review · Reviewer_cXkh · 2023-07-09

**Soundness:** 3 good
**Presentation:** 3 good
**Contribution:** 2 fair
**Rating:** 7
**Confidence:** 4

**Summary:**

This manuscript presents a model-agnostic solution for segmentation refinement, which implements the refinement process of the coarse masks through a series of denoising diffusion steps. Comprehensive experiments on various segmentation tasks are conducted, including semantic segmentation, instance segmentation, and dichotomous image segmentation and the results demonstrate the superiority of the proposed method.

**Strengths:**

- They obtain SOTA results on multiple tasks.
- The idea sounds quite interesting.
- This manuscript is well-formulated and -presented.

**Weaknesses:**

The motivation behind lacks focus. The manuscript primarily discusses the problem of the existing methods of segmentation refinement in Introduction but fails to indicate the reason why denoising diffusion models are suitable and important for the task of segmentation refinement. What are the technical challenges in applying denoising diffusion models to segmentation refinement? To address these challenges, what innovations does this paper propose regarding the application of denoising diffusion models to segmentation refinement?

How is the value of the state transition probability specifically calculated?

There are several instances where the use of others' work, such as Gumbel-max sampling, has not been properly cited.

For the DIS experiment, commonly used evaluation metrics for DIS are not employed.

Some grammar and spelling errors.

**Questions:**

See above comments.

**Limitations:**

As mentioned in the conclusion, the diffusion process seems to lead to a slowdown in inference due to the multi-step iterative strategy. As a post-processing step for segmentation tasks, this is indeed a major drawback. Did the authors conduct any experiments to examine the relationship between iteration step, accuracy, and speed?

---

> ### Author Rebuttal · Authors · 2023-08-10
>
> # Response to Reviewer cXkh
> ### **Q1. (a) Why using diffusion**
> As mentioned in Line 40 and Fig. 1 in the paper, we found there are diverse types of errors presented in the coarse masks. This is the reason for the underperformance of existing methods because attempting to address all these diverse errors at once is extremely challenging. Therefore, a very natural idea is: **since correcting all errors simultaneously is too difficult, can we make the model correct only some "most obvious errors" at each step and iteratively converge to an accurate result?**
>
> Regarding this idea, we found that this is precisely how the denoise diffusion models tackle the challenges in generative tasks: diffusion models decompose the fitting process from Gaussian noise to data samples into multiple time steps, making it easier and more accurately to fit the distribution of data samples. Therefore, transferring the diffusion model to refinement task is a very natural way to achieve the aforementioned objective.
>
> ### **Q1. (b) Technical challenges and innovations in applying denoising diffusion models**
> There are two main challenges in applying denoising diffusion models to segmentation refinement task:
> * Firstly, most existing diffusion models adopt the Gaussian assumption, representing the forward diffusion process as adding Gaussian noise, which is not suitable for binary segmentation predictions.
> * Secondly, nearly all existing diffusion frameworks have employed random noise as the origination for sampling, whereas we aim to initiate sampling from the coarse mask, which is a binary foreground vs. background map.
>
> Given these two challenges, The proposed method introduces the following innovations:
> * We define every pixel in the segmentation results to have two states, corresponding to fine segmentation (ground truth) and coarse segmentation. As a result, the forward and backward processes of diffusion are naturally represented as stochastic states-transition, solving the first challenge.
> * We formulate the states-transition as a unidirectional process, thus ensuring its convergence to the coarse mask during forward process, addressing the second challenge.
> * Furthermore, in the reverse diffusion process, we devised a confidence-based states-transition strategy (as shown in Eq. (11) in the paper), which ensures that pixels with the highest confidence predictions are updated first. This naturally achieves the objective of refining "most obvious errors" at each step, as mentioned in Q1.(a).
>
>
> ### **Q2. Value of the states-transition probability**
> The states-transition probability of forward process is defined in Eq. (6), which is determined by hyperparameter $\beta$. In the reverse process, states-transition probability is defined in Eq. (11), which is derived from Eq. (9) by substituting $x_0$ with $p_\theta(\tilde m_{0|t})^{i,j}$ as mentioned in Line 162 in the paper.
>
> ### **Q3. Citation, grammar and spelling problem**
> Thank you for your valuable and thoughtful comments. We will carefully check the citation and English writing throughout the entire paper and make corrections in the revision.
>
> ### **Q4. DIS evaluation metrics**
> In the main paper, we chose IoU and mBA as metrics for the DIS experiment because these two metrics are the most intuitive in reflecting mask quality and accuracy. Regarding your concern, we also report the 5 metrics used in the original paper of DIS, as shown in **Tab. 1 in the rebuttal PDF file of General Response**. Experimental results demonstrate that our method consistently improves all metrics, indicating its effectiveness in improving segmentation accuracy and achieving high-quality mask refinement under different focuses.
>
> ### **Q5. The relationship between iteration step, accuracy, and speed**
> We conduct the following experiment on instance segmentation about the relationship between the model's accuracy, computational complexity, time consumption (the average time consumed per image), and the number of iteration steps. Results are shown in the table below. It can be observed that both time consumption and computational complexity increase linearly with the number of steps, while accuracy reaches saturation around 5-6 steps.
>
> | Steps       |  0   |  1   |  2   |  3   |  4   |  5   |  6   |  7   |  8   |
> |:----------- |:----:|:----:|:----:|:----:|:----:|:----:|:----:|:----:|:----:|
> | Mask AP     | 39.8 | 40.7 | 41.0 | 41.4 | 41.6 | 41.9 | 41.9 | 41.9 | 41.9 |
> | Boundary AP | 27.3 | 29.9 | 30.6 | 31.5 | 32.1 | 32.5 | 32.6 | 32.6 | 32.6 |
> | Time(s)     | n/a  | 0.52 | 1.01 | 1.53 | 1.98 | 2.45 | 2.94 | 3.44 | 3.92 |
> | FLOPs(G)    | n/a  | 249  | 497  | 746  | 994  | 1243 | 1492 | 1740 | 1989 |
>
> We also provide a comparison between our method and existing methods in the following table. For semantic segmentation:
>
> | Method     | Time(s) | FLOPs(G) | Params(M) |
> | ---------- |:-------:|:--------:|:---------:|
> | CascadePSP |   6.2   |  26518   |   67.62   |
> | CRM        |   4.3   |   2356   |   9.27    |
> | **Ours**   |   1.6   |   1492   |  118.93   |
>
> For instance segmentation:
>
> | Method   | Time(s) | FLOPs(G) | Params(M) |
> | -------- |:-------:|:--------:|:---------:|
> | BPR      |   1.4   |   135    |   65.85   |
> | **Ours** |   2.9   |   1492   |  118.93   |
>
> Results indicate that, in semantic segmentation, our method demonstrates faster speed and lower computational complexity. However, in instance segmentation, our method is slower than previous approaches.
>
> This is largely due to the high computational complexity of the U-Net model used in previous diffusion-based works (with 249 GFLOPs per step). In this work, we focuses on exploring the feasibility and superiority of diffusion process in the segmentation refinement task, thus we maintain the model structure unchanged. In the future, we will continue to explore more efficient segmentation refinement by replacing the U-Net used in our method with a more lightweight architecture.

---

> > ### Comment · Reviewer_cXkh · 2023-08-10
> >
> > Thank you for your impressive feedback. The majority of my concerns have been addressed. However, I still have a few remaining questions about motivation that I would like to discuss further.
> >
> > Regarding your response to Q1 (a):
> > - What are the advantages and insights of using diffusion for segmentation, and further for high-accuracy segmentation?
> >
> > Regarding your response to Q1 (b):
> > - Why is the addition of Gaussian noise to the diffusion process deemed unsuitable for binary segmentation tasks?
> > - How does the proposed technique demonstrate generazability to both general segmentation tasks (such as instance/semantic segmentation) and high-accuracy segmentation tasks (like DIS)? What differentiates the usage of SegRefiner between these two task types? Does this imply that SegRefiner can be used for ultra-accuracy segmentation tasks, akin to alpha matting datasets?
> >
> > Furthermore, I would like to inquire whether the authors have uploaded the revised version of the paper (*.pdf) to the system.

---

> > > ### Author Response · Authors · 2023-08-10
> > > **Response to Reviewer cXkh**
> > >
> > > ### **Q1. Advantages of using diffusion for segmentation**
> > > The advantages of diffusion model are mainly manifested in the following aspects:
> > > * Firstly, the diffusion model exhibits a strong perceptual capability for the texture and edge details of images. This conclusion has been demonstrated in numerous diffusion-based image generation works since diffusion models can generate samples with extremely realistic details. The reason for this might lie in the fact that, as a latent variable generative model, diffusion's latent variables are full-sized. Similar to the findings in StyleGAN \[A\], larger-sized latent variables tend to be more inclined towards controlling details and textures. This might explain the perceptual sensitivity of diffusion towards details. Thus, such a detail-sensitive model is particularly suitable for pixel-level segmentation task (especially for high-accuracy segmentation which emphasizes capturing very fine details).
> > > * Secondly, as mentioned in the previous response, addressing all errors in a single iteration can be exceptionally challenging. In contrast, the iterative strategy of diffusion model reduces the difficulty of segmentation refinement tasks by making the model focus on only some "most obvious errors" (subset of all errors) during each step/iteration. This iterative procedure gradually refines the segmentation mask, leading to improved accuracy. In image generation, this iterative strategy imparts the diffusion model with the ability to generate realistic images of challenging complex scenes. When applied to segmentation refinement task, it empowers our SegRefiner to handle more challenging examples and consistently rectify errors and produce precise predictions.
> > >
> > > \[A\] Karras, Tero, Samuli Laine, and Timo Aila. "A style-based generator architecture for generative adversarial networks." in CVPR 2019.
> > >
> > > ### **Q2. Why is the addition of Gaussian noise unsuitable**
> > > * Firstly, the reason for adding Gaussian noise in image generation fundamentally stems from the assumption that natural images, as target data, can be considered as high-dimensional Gaussian variables. Thus, the forward and backward processes are interpreted as noising and denoising and the core idea is to fit a Gaussian distribution. However, in this perspective, our target data is binary masks, which obviously do not conform to the Gaussian distribution. Therefore, we should not regard the process of obtaining segmentation masks as fitting a Gaussian variable. On the contrary, representing it with a discrete random states-transition process is very natural.
> > > * Secondly, as a segmentation refinement method, our core idea is to treat the coarse prediction as a noisy ground truth and recover the high-quality segmentation by eliminating this noise. Since the coarse segmentation result entails misclassifications during foreground/background prediction, it is more akin to discrete states transitions occurring in certain pixels of the ground truth, rather than a binary ground truth overlayed with a continuous Gaussian noise.
> > >
> > > ### **Q3. Generazability of SegRefiner**
> > > * The generazability largely comes from the model-agnostic and class-agnostic manners of SegRefiner. Its mechanism is first obtaining prompts from the coarse mask to determine the objects needed to be segmented and then providing accurate masks. Such a mechanism allows SegRefiner to have consistent objectives across different segmentation tasks, enhancing the generazability of SegRefiner.
> > > * The difference lies in the fact that for DIS, the dataset resolution is exceptionally high, while the model's output size is only $256 \times 256$. This resolution is inadequate for such high-resolution datasets and results in the loss of numerous fine details. Therefore, we employed a combination of both global and local refinements strategy (as elaborated in line 217 of the paper). However, for instance segmentation on the LVIS dataset with relatively smaller resolution, we only employed instance-level refinement without local refinement.
> > > * The alpha matting task requires predicting floating-point values for unknown region. Although our method is designed for binary masks using a discrete process, it can also handle image matting by converting the intermediate predicted fine mask into a floating-point format. However, since this approach would require the aforementioned modifications and training on matting datasets independently, resulting in substantial deviations from the overall framework of this work, we did not conduct the matting experiment within the scope of this work.
> > >
> > > ### **Q4. "whether the authors have uploaded the revised version of the paper (*.pdf) to the system."**
> > > According to the author responses policy of NeurIPS 2023, *"Authors may not submit revisions of their paper or supplemental material"*, we have not uploaded the revised vision. We will update the paper according to the reviewers' comments. Thank you very much for your valuable comments!

---

> > > > ### Comment · Reviewer_cXkh · 2023-08-10
> > > >
> > > > Thanks for your illustrations. It would be nice when you corporate these discussion into your paper or supplementary materials, just for better understanding for readers. After discussion, I am happy to change my rating to acceptance. Looking forward to the final publication.
> > > >
> > > > Best.

---

> > > > > ### Author Response · Authors · 2023-08-10
> > > > > **Thanks for the Response of Reviewer cXkh**
> > > > >
> > > > > Dear Reviewer cXkh,
> > > > >
> > > > > Thank you very much for your careful response and for increasing the rating to acceptance.
> > > > >
> > > > > We will add these discussions to the main paper or supplementary materials in the revision following your valuable suggestions.
> > > > >
> > > > > Best regards,
> > > > > Authors

---

### Official Review · Reviewer_ygQQ · 2023-07-11

**Soundness:** 4 excellent
**Presentation:** 3 good
**Contribution:** 4 excellent
**Rating:** 8
**Confidence:** 4

**Summary:**

This work proposes diffusion-based image segmentation refinement approach, SegRefiner, using a newly designed discrete diffusion process. SegRefiner performs model-agnostic segmentation refinement that takes in coarse output masks from any segmentation model as an input and refines them through a series of diffusion steps while estimating state-transition probabilities for each pixel. Comprehensive experiments across multiple segmentation tasks demonstrates that SegRefiner consistently improves segmentation metrics and outperforms state-of-the-art model-specific and model-agnostic refinement approaches.

**Strengths:**

Originality
This work has multiple novel contribution ranging from - being first to introduce diffusion-based image segmentation refinement approach, the use of discrete diffusion instead of Gaussian diffusion process of DDPM, being class-agnostic as well as ability to capture extremely fine-details when applied to high-resolution images.

Quality
The paper is well-written and mostly easy to follow. Authors have put their best effort to condense lot of details into conference paper length. As the use of discrete diffusion is new in practice, some of the details can be elaborated using examples in the supplementary for the ease audience.

Significance
The approach taken by SegRefiner is unique. For instance, mask and states prediction task is kept separate. Similarly, there is lot of subtle details required to run this successfully as listed under the training strategy, noise schedule and other settings. All these contribution can be quite useful to other researcher pursuing different applications using generative model.

**Weaknesses:**

None that are significant. Have listed one under the limitation section.

**Questions:**

1. How is the posteior in eq(9) derived ? Similarly how do one derive eq(11) ? And hence where is eq(9) used ?
2. Is the refinement of semantic and instance segmentation performed using zero-shot ? All I understand is LR- and HR-SegRefiner are trained using LVIS and composite dataset respectively.

**Limitations:**

Yes, authors have address one of the core limitation to using diffusion process. As far as my understanding, another limitation of such class-agnostic and model-agnostic approach is one should undertake each instance one at a time for refinement. This probably due to the fact there is no class information input to SegRefiner model.

---

> ### Author Rebuttal · Authors · 2023-08-10
>
> # Response to Reviewer ygQQ
> ### **Q1. About Eq. (9) and Eq. (11)**
> **Firstly, for Eq. (9),** it's derived from the Bayes' theorem：
> \begin{equation}\begin{split}
> q(x_{t-1}|x_t, x_0) = \frac{q(x_t|x_{t-1}, x_0)q(x_{t-1}|x_0)}{q(x_t|x_0)}.
> \end{split}\end{equation}
>
> According to Eq. (6) and Eq. (7), we can get all the items in Bayes' theorem as:
> \begin{equation}\begin{split}
> q(x_t|x_{t-1}, x_0) &= x_{t-1} Q_t x_t^T, \\\\
> q(x_{t-1}|x_0)) &= x_{0} \bar Q_{t-1} x_{t-1}^T, \\\\
> q(x_{t}|x_0)) &= x_{0} \bar Q_{t} x_{t}^T.
> \end{split}\end{equation}
> It appears to be slightly different from Eq. (6). The reason is that the results of Eq. (6) is a length-2 vector, representing the probabilities of two states respectively. While we use a scalar form here, which includes an additional one-hot vector $x_*^T$ and is more suitable for subsequent derivations. For the denominator of Eq. (9), we have obtained the same form. Let's consider the numerator:
> \begin{equation}\begin{split}
> q(x_t|x_{t-1}, x_0)q(x_{t-1}|x_0) &= x_{t-1} Q_t x_t^T \cdot x_0 \bar Q_{t-1} x_{t-1}^T \\\\
> &= x_{t-1} (x_t Q_t^T)^T \cdot x_0 \bar Q_{t-1} x_{t-1}^T \\\\
> &= (x_t Q_t^T)x_{t-1}^T \cdot (x_0 \bar Q_{t-1}) x_{t-1}^T \\\\
> &= \[(x_t Q_t^T) \odot (x_0 \bar Q_{t-1})\] x_{t-1}^T ,
> \end{split}\end{equation}
> where "$\cdot$" refers to scalar multiplication and "$\odot$" refers to element-wise multiplication. This is the scalar form, by removing the $x_{t-1}^T$, we can get the corresponding vector form just as Eq. (9).
>
> **Secondly, for Eq. (11),** it's derived from Eq. (9) by substituting $x_0$ with $[p_0, 1-p_0]$ ($p_0$ refers to $p_\theta(\tilde m_{0|t})^{i,j}$ in Line 162 in the paper for simplicity). Let's start from the scalar form above:
> \begin{equation}\begin{split}
> p(x_{t-1}|x_{t})
> &= \frac{\[(x_t Q_t^T) \odot (x_0 \bar Q_{t-1})\] x_{t-1}^T}{x_{0} \bar Q_{t} x_{t}^T} \quad | \quad x_0=\[p_0, 1-p_0\] \\\\
> &= \frac{x_t (Q_t^T \odot \[p_0 \bar \beta_{t-1}, 1 - p_0 \bar \beta_{t-1}\] ) x_{t-1}^T}{\[p_0 \bar \beta_{t}, 1 - p_0 \bar \beta_{t}\] x_t^T} \\\\
> &= \frac{x_t \begin{bmatrix}
>     p_0 \bar \beta_t & 0 \\\\
>     p_0 (\bar \beta_{t-1} - \bar \beta_t) & 1 - p_0 \bar \beta_{t-1}
>     \end{bmatrix} x_{t-1}^T}{x_t \[p_0 \bar \beta_{t}, 1 - p_0 \bar \beta_{t}\]^T} \\\\
> &= x_t \begin{bmatrix}
>    1 & 0 \\\\
>     \frac {p_{0} (\bar \beta_{t-1} - \bar \beta_t)}{1 - p_0 \bar \beta_{t}} & \frac{1 - p_0 \bar \beta_{t-1}}{1 - p_0 \bar \beta_{t}} \\\\
>     \end{bmatrix} x_{t-1}^T.
> \end{split}\end{equation}
> Same as Eq. (9), we can remove $x_{t-1}^T$ and get the vector form of Eq.(11). Therefore, Eq. (9) is used to obtain Eq. (11). This follows the same logic of DDPM, which substitutes $x_0$ in the posterior with a predicted one and gets the reversed probability distribution.
>
> ### **Q2. Whether zero-shot or not**
> The proposed method is class-agnostic and model-agnostic, offering the advantage of being applied in a zero-shot manner. The reported experiments, while indicative of the approach's effectiveness, do not constitute a fully rigorous zero-shot evaluation:
> * Experiment of instance segmentation is conducted on the LVIS validation set, the training set of which is used to train the LR-SegRefiner, thus it's not zero-shot.
> * For semantic segmentation, we use the HR-SegRefiner and evaluate it on the BIG dataset. It indeed involves different training and testing datasets. While many categories in BIG dataset are also presented in the composite training data. Thus it's not a completely rigorous zero-shot experiment.
>
> To evaluate the zero-shot capability of our approach, we report results on the FSS-1000 dataset \[A\], which is comprised of 1000 uncommon categories. The results on 4 unseen/novel categories (column 2 to column 5) and across the entire dataset（column 6）are presented in the table below, where the coarse masks are derived from the baseline of \[A\]. It can be observed that our approach achieves consistent improvements in a zero-shot manner on these uncommon categories.
>
> | Category                       | Rice cooker | Nintendo Switch | Windmill | Hot water heater | All categories |
> |:------------------------------ |:-----------:|:---------------:|:--------:|:----------------:|:-----------:|
> | Baseline              |    91.1     |      80.5       |   90.3   |       83.8       |    91.1     |
> | &nbsp;&nbsp;&nbsp;&nbsp;+ Ours |    96.3     |      94.2       |   91.7   |       87.1       |    93.4     |
> | Improvement                    |     +5.2     |      +13.7       |   +1.4    |       +3.3        |     +2.2     |
>
> \[A\] Li, Xiang, et al. "FSS-1000: A 1000-Class Dataset for Few-Shot Segmentation." CVPR'20.
>
>
> ### **Q3. Only one instance at a time**
> Thank you for your insightful comments. It's truly a limitation that our method can only refine one mask at a time. While it's not due to its class-agnostic manner. Instead, it is constrained by two factors:
> * Firstly, refining multiple masks at once would require taking the entire image region as input, resulting in a smaller-scale output for each mask and the loss of fine-grained information.
> * Secondly, even if we tolerate the information loss, a larger challenge arises from the widespread occurrence of overlap between multiple coarse masks, which means we would need to establish a multiple-input-output parallel framework. While, without significant modifications to the model architecture, this is not significantly different from independently refining multiple masks and distinguishing them in a batch.
>
> Since this work focuses on exploring the feasibility of diffusion in the refinement task and the superiority of our diffusion process, we have chosen the simplest approach: maintaining the U-Net architecture used in previous diffusion-based works unchanged and refining every mask independently. It's an important direction we will continue to explore in the future to design an efficient parallel model architecture to enable our method to handle multiple masks at once.

---

### Official Review · Reviewer_cPQx · 2023-07-26

**Soundness:** 4 excellent
**Presentation:** 3 good
**Contribution:** 3 good
**Rating:** 7
**Confidence:** 3

**Summary:**

This paper proposes a novel method for improving the quality of object masks generated by different segmentation models. The method, called SegRefiner, is model-agnostic, meaning that it can be used with different segmentation models. SegRefiner works by interpreting segmentation refinement as a data generation process. This allows the work to implement the refinement process as a series of denoising diffusion steps.

Specifically, SegRefiner takes coarse masks as input and refines them using a discrete diffusion process. For each pixel, it predicts the label and corresponding state-transition probabilities. This allows the model to progressively refine the noisy masks in a conditional denoising manner.

The method is evaluated on a variety of segmentation tasks, and the results show that SegRefiner consistently outperforms previous methods. It improves both the segmentation metrics and boundary metrics across different types of coarse masks. SegRefiner also exhibits a strong capability to capture extremely fine details when refining high-resolution images.

**Strengths:**

1. Overall, the paper is well written and clearly organized.

2. The method is intuitive and straightforward. The performance improvements are clear compared to the previous baselines.

3. The inference is efficient for diffusion as it only uses 6 timesteps.


**Weaknesses:**

1. Not necessarily a weakness but the model does not clearly state how the coarse masks are generated for training until later sections. While the proposed method follows previous works on this part, it would still improve the clarity if this is discussed in earlier sections and in the overview figures.

2. The paper only shows one example of how each step of the diffusion process works during inference. The authors are recommended to include more examples with some highlighted boundary comparisons.


**Questions:**

Please see the weakness section.

**Limitations:**

The authors properly addressed the limitations in the paper.

---

> ### Author Rebuttal · Authors · 2023-08-10
>
> # Response to Reviewer cPQx
> ### **Q1. The coarse masks used for training**
> We very appreciate your insightful comments and will clarify how the coarse masks are generated in earlier sections and in the overview figures in our next version.
>
> ### **Q2. More examples of inference**
> More additional visualizations of the inference process are provided in Fig. 1 in the rebuttal PDF file of General Response. We will include these qualitative results in our next version to help readers gain a more intuitive understanding of this work.

---

> > ### Comment · Reviewer_cPQx · 2023-08-18
> >
> > Thanks for preparing the rebuttal! I don't have other concerns that need to be addressed.

---

### Author Rebuttal · Authors · 2023-08-10

## General Response
We have uploaded a PDF file here, which includes:
* **Table 1:** (to Reviewer cXkh Q4) An additional experiment to evaluate SegRefiner on DIS-5K dataset with the commonly used metrics for Dichotomous Image Segmentation.
* **Figure 1:** (to Reviewer cPQx Q2) Visualizations of how each step of SegRefiner works during inference.
* **Figure 2:** (to Reviewer oFWK Q2) An example to illustrate why Boundary AP improves more in refinement of instance segmentation.

---

### Decision · Program_Chairs · 2023-09-21

**Decision:**

Accept (poster)

**Comment:**

The submission was reviewed by 5 reviewers. All reviewers agree that the ideas presented in the paper are interesting and find the approach to sound. Moreover, the reviewers complement the presentation of the paper and find the validation convincing. AC agrees with the reviewers and recommend to accept the submission.